# Single-cell profiling reveals the intratumor heterogeneity and immunosuppressive microenvironment in cervical adenocarcinoma

Yang Peng[1†], Jing Yang[2†], Jixing Ao[3], Yilin Li[4], Jia Shen[5,6,7], Xiang He[5,6,7], Dihong Tang[1], Chaonan Chu[1], Congrong Liu[2]*, Liang Weng[2,5,6,7]*

[1]Fourth Department of Gynecologic Oncology, Hunan Cancer Hospital, The Affiliated Cancer Hospital of Xiangya School of Medicine, Central South University, Changsha, China; [2]Department of Pathology, Third Hospital, School of Basic Medical Sciences, Peking University Health Science Center, Beijing, China; [3]Department of Gynecologic Oncology, Changsha Kexin Cancer Hospital, Changsha, China; [4]Department of Pathology, Hunan Cancer Hospital/The Affiliated Cancer Hospital of Xiangya School of Medicine, Central South University, Changsha, China; [5]Xiangya Cancer Center, Xiangya Hospital, Central South University, Changsha, China; [6]Hunan International Science and Technology Collaboration Base of Precision Medicine for Cancer, Changsha, China; [7]Key Laboratory of Molecular Radiation Oncology of Hunan Province, Changsha, China

*For correspondence:
congrong_liu@hsc.pku.edu.cn (CL);
wengliang@pku.edu.cn (LW)

[†]These authors contributed equally to this work

Competing interest: The authors declare that no competing interests exist.

## eLife Assessment

In this **useful** article, the authors performed scRNA-seq on a diverse cohort of 15 early-stage cervical cancer patients. Correlative data is provided to support the possible establishment of an immunosuppressive microenvironment near SCL26A3+ cells, and an association of these cells with upstaging at time of surgery. However without more extensive validation, the evidence supporting the conclusions remains **incomplete**. Overall, this article will provide a potentially helpful dataset for researchers studying cervical cancer.

## Abstract

**Background:** Cervical adenocarcinoma (ADC) is more aggressive compared to other types of cervical cancer (CC), such as squamous cell carcinoma (SCC). The tumor immune microenvironment (TIME) and tumor heterogeneity are recognized as pivotal factors in cancer progression and therapy. However, the disparities in TIME and heterogeneity between ADC and SCC are poorly understood.
**Methods:** We performed single-cell RNA sequencing on 11 samples of ADC tumor tissues, with other 4 SCC samples served as controls. The immunochemistry and multiplexed immunofluorescence were conducted to validate our findings.
**Results:** Compared to SCC, ADC exhibited unique enrichments in several sub-clusters of epithelial cells with elevated stemness and hyper-malignant features, including the Epi_10_CYSTM1 cluster. ADC displayed a highly immunosuppressive environment characterized by the enrichment of regulatory T cells (Tregs) and tumor-promoting neutrophils. The Epi_10_CYSTM1 cluster recruits Tregs via ALCAM-CD6 signaling, while Tregs reciprocally induce stemness in the Epi_10_CYSTM1 cluster through TGFβ signaling. Importantly, our study revealed that the

Epi_10_CYSTM1 cluster could serve as a valuable predictor of lymph node metastasis for CC patients.

**Conclusions:** This study highlights the significance of ADC-specific cell clusters in establishing a highly immunosuppressive microenvironment, ultimately contributing to the heightened aggressiveness and poorer prognosis of ADC compared to SCC.

**Funding:** Funded by the National Natural Science Foundation of China (82002753; 82072882; 81500475) and the Natural Science Foundation of Hunan Province (2021JJ40324; 2022JJ70103).

## Introduction

Cervical cancer (CC) is the fourth leading cause of cancer-related death among all types of women malignancies (*Sung et al., 2021*). The two major histological types of CC are squamous cell carcinoma of cervix (SCC) and adenocarcinoma of cervix (ADC), accounting for 70% and 25% of all cases, respectively (*Cohen et al., 2019*). Compared to SCC, ADC patients have a higher rate of rapid progression, relapse, and insensitivity to chemotherapy, radiotherapy, and even immunotherapy. Therefore, ADC is considered a more aggressive phenotype than SCC (*Sasieni et al., 2009*). Unfortunately, the current standard treatment for CC is not stratified based on pathologic types. As a result, ADC patients often have suffered from therapeutic failures and experienced short-term recurrence and metastasis, even finishing standard treatment procedures (*Sasieni et al., 2009*). However, due to the low prevalence of ADC, in-depth investigations are insufficient. Furthermore, the molecular mechanisms elucidating its aggressiveness and the associated biomarkers are not yet well understood. Therefore, this study aims to comprehensively compare the intratumor heterogeneity of ADC, hoping to find out the potential molecule markers specifically associated with ADC.

For early-stage CC patients, it is a crucial and unavoidable issue to have post-surgical upstaging. Over 10% of patients, who are initially diagnosed with early stage and deemed suitable for radical surgery, end up being upstaged due to lymph node (LN) metastasis discovered after the surgery (*Dabi et al., 2018*; *Thelissen et al., 2022*). This indicates misdiagnosis, leading to different therapeutic strategies and outcome. According to the National Comprehensive Cancer Network (NCCN) guidelines, these misdiagnosed patients require additional radiotherapy after surgery, even though they have underwent surgery (*Bhatla et al., 2018*). However, combo treatment of surgery and radiotherapy may cause additional complications and heavier financial burdens, without offering more benefits than radiotherapy alone (*Cushman et al., 2018*; *Kashima et al., 2023*). Actually, the optimal therapeutic approach should be radiotherapy alone if the presence of LN metastasis could have been accurately diagnosed beforehand (*Abu-Rustum et al., 2020*). To avoid this problem, the International Federation of Gynecology and Obstetrics (FIGO) in 2018 updated a new stage as IIIC by including LN status and stratified this stage into two subgroups: IIICp (p stands for pathological findings of LN, which means misdiagnosis) and IIICr (r stands for radiological findings of LN, which means pre-surgical evidence) (*Bhatla et al., 2018*). Unfortunately, radiological imaging tools do have limitations since CT or MRI can only detect suspiciously metastatic LN when the minimum diameter is ≥10 mm (*Saleh et al., 2020*). Therefore, together with radiological diagnosis, it is urgent to search for potential molecular predictors of LN metastasis in order to decrease the rate of post-surgical upstaging.

Persistent infection with human papillomavirus (HPV) is the primary cause of CC (*Mann et al., 2023*). However, there are about 5–15% of patients testing negative for HPV (*Yang et al., 2023*). It is reported that HPV-negative patients tend to experience worse clinical outcomes (*Fernandes et al., 2022*). However, current therapeutic strategies for CC do not take into account the HPV infection status as a factor for stratified therapy. Therefore, it is imperative to investigate the precise mechanisms underlying the development of HPV non-infected tumors.

The tumor immune microenvironment (TIME), consisting of tumor cells, immune cells, cytokines, and their interactions, predominantly determines the efficacy of immunotherapy (*Lv et al., 2022*). In CC, while it has been reported that 30–50% of patients exhibit positive PD-L1 expression, however, only an estimated 20–30% are expected to respond positively to PD-1/PD-L1 blockade therapy (*Omenai et al., 2022*; *Sun et al., 2020*). Therefore, CC, especially the pathological type of ADC, is characterized as an immune-insensitive cancer type due to atypical modulation of the TIME (*Cao et al., 2023*). The mechanisms of tumor immune evasion are basically ascribed to dysregulation of TIME and are similar across different types of cancer, involving the deactivation of immune surveillance pathways or

recruitment of immunosuppressive cells (*Vinay et al., 2015*). However, the specific characteristics of the TIME in ADC remain poorly understood. Therefore, in order to optimize precise therapeutic strategies and enhance the efficacy of immunotherapy, it is indeed imperious to investigate the intricacies of TIME in ADC.

To extensively investigate the cellular and molecular heterogeneity of CC, particularly ADC, the utilization of single-cell RNA sequencing (scRNA-seq) technique is crucially needed. In the current study, we focused on the remodeling of TIME in ADC and performed scRNA-seq analysis on 11 ADC samples, with another 4 SCC samples included as controls. By high-throughput analysis, we are able to characterize the landscape of TIME in ADC, with different types of tumor-promoting cells being recruited in the vicinity of the tumor, thus creating an immunosuppressive environment in ADC. The cellular crosstalk analysis provides valuable insights into the interactions between different cell types within the TIME. Furthermore, we have conducted a screening of novel signature genes as prognostic biomarkers for ADC. These biomarkers have the potential to possibly predict LN metastasis and may guide personalized treatment decisions more effectively.

## Results

### Single-cell transcriptomic landscape of cervical cancer

We collected 15 samples of CC tissues from 15 individual patients immediately after undergoing radical hysterectomy surgery. Among these cases, 11 were ADC types, with 5 being HPV-negative and 6 being HPV-positive. The remaining four cases were SCC types, with an even distribution of HPV-positive and HPV-negative status (*Supplementary file 1*). Prior to surgery, all 15 patients were diagnosed as early FIGO stages (stage I–IIA) based on evaluation of CT, MRI, and physical examination. However, after surgery, five cases turned out to be upstaged with pathological evidence of LN metastasis (i.e., FIGO stage IIICP), requiring theses patients to receive radiation therapy according to NCCN guidelines. Therefore, in addition to investigate the difference between histological types and HPV infection status, we also focused on the issue of post-surgical upstaging by comparing different FIGO stages.

In order to analyze the genomic profiling, we resolved these samples at the single-cell level using the 10X Genomics chromium platform. The cell viability and sample quality were ensured beforehand, and a total of 102,012 qualified cells were utilized for further analysis (*Figure 1A*). Uniform Manifold Approximation and Projection (UMAP) was utilized to reduce the dimension of gene profiling data for visualization. The obtained cells were primarily categorized into 12 cell clusters, which were further re-clustered into 9 distinct cell types, based on specific gene markers (*Li et al., 2022*; *Figure 1B and C*). These cell types included T cells (38,627 cells in total, proportioning 37.87%, marked with PTPRC, CD3D, CD3E, and CD3G), epithelial cells (32,199 cells in total, proportioning 31.56%, marked with EPCAM, SLP1, and CD24), neutrophils (12,586 cells in total, proportioning 12.34%, marked with CSF3R), macrophages (6798 cells in total, proportioning 12.34%, marked with CD68 and CD163), fibroblasts (4258 cells in total, proportioning 4.17%, marked with COL1A2 and DCN), plasma cells (4044 cells in total, proportioning 3.96%, marked with JCHAIN), endothelial cells (1389 cells in total, proportioning 1.36%, marked with ENG and VWF), B cells (1326 cells in total, proportioning 1.30%, marked with CD79A and MS4A1), and mast cells (785 cells in total, proportioning 0.77%, marked with MS4A2) (*Figure 1D and E* and *Supplementary files 2* and *3*).

According to our scRNA-seq data, ADC exhibited a significant higher proportion of neutrophils than SCC (20.05% vs 4.23%). The proportion of T cells was also higher in ADC (41.13%) compared to SCC (35.48%). Conversely, SCC showed a nearly threefold enrichment of epithelial cell (47.43%) compared to ADC (16.48%) (*Figure 1F*). When considering the HPV infection status, HPV-negative patients demonstrated decreased proportions of epithelial cells (7.78% vs 28.38%) and neutrophils (16.66% vs 24.68%) compared to HPV-positive patients. The proportion of T cells enriched in HPV-negative cases (45.77%) was higher than that in HPV-positive cases (32.42%). HPV-negative patients exhibited prominently higher proportions of plasma cells (10.13% vs 1.99%) and B cells (2.84% vs 0.77%) compared to HPV-positive patients, even though their proportions were lower than T cells, epithelial cells, and neutrophils (*Figure 1—figure supplement 1A*). Furthermore, we conducted a comparison between early-stage and late-stage cases, revealing that late-stage patients have a higher proportion of epithelial cells (35.50%) than early-stage cases (28.82%) (*Figure 1—figure supplement*

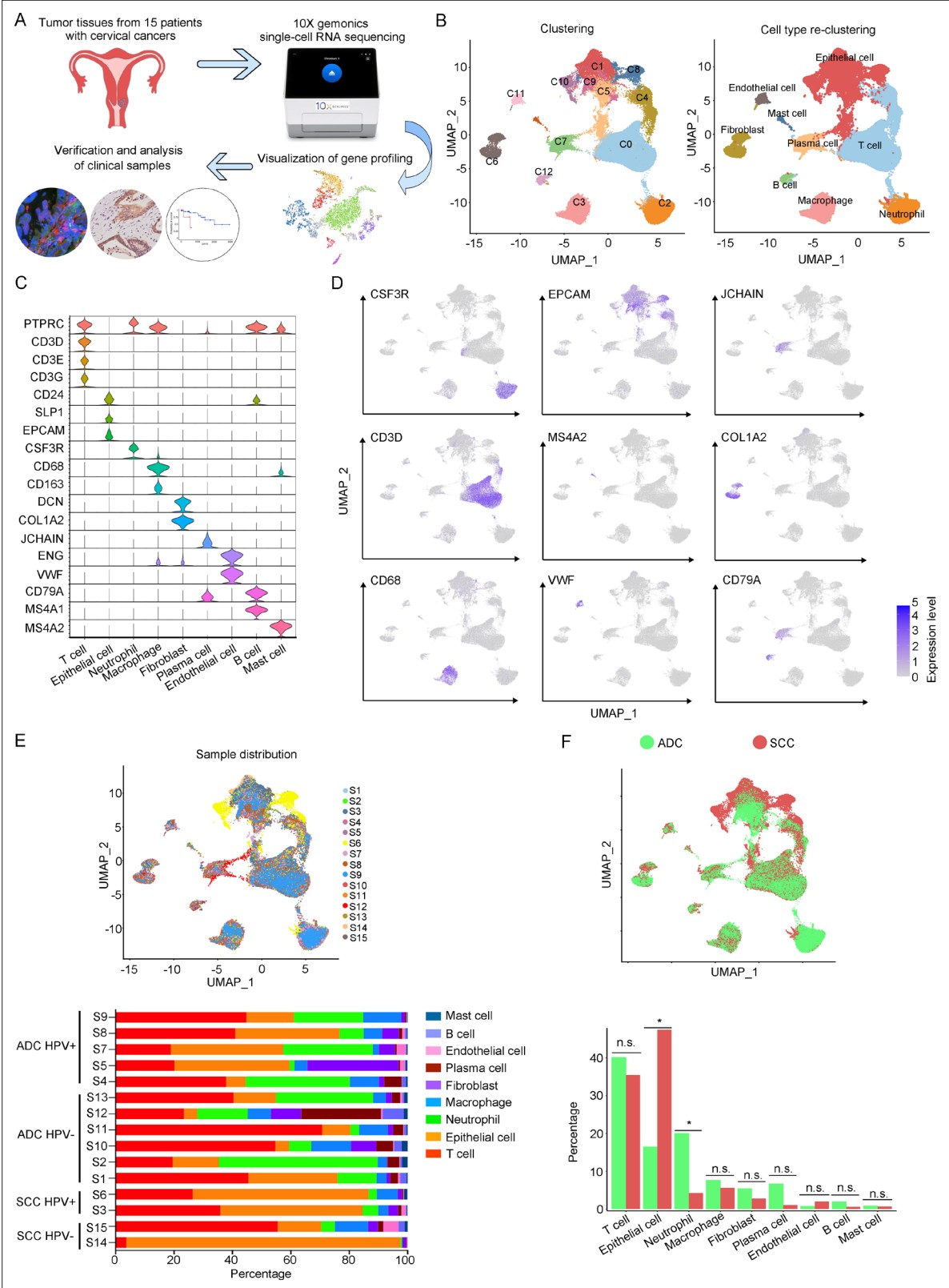

**Figure 1.** The single-cell genomic atlas of cervical cancer. (**A**) The schematic design of sample collection, single-cell RNA sequencing, data processing, and clinical validating through 10X Genomics platform (the sketch of scRNA-sq machine, as well as the plot of dimension-reduction visualization, was originally cited from the official website of 10X Genomics: https://www.10xgenomics.com). (**B**) Uniform Manifold Approximation and Projection (UMAP) plots of all single cells by original clustering and cell type reclustering, according to the published lineage-specific marker genes, respectively. (**C**)

*Figure 1 continued on next page*

*Figure 1 continued*

Violin plots demonstrating the expression of marker genes that correspond to each of the major cell types. (**D**) UMAP plots demonstrating the most typical marker gene specific to each type of cell cluster. (**E**) UMAP plot and histogram plot showing the occupation ratios of the major cell types in each individual sample, with human papillomavirus (HPV) status and histological type summarized. (**F**) UMAP (up) and histogram (down) plots to show the differences of distribution and proportion of each cell type between different histological types (adenocarcinoma [ADC] vs. squamous cell carcinoma [SCC]). Statistics were performed using R software with two-sided Wilcoxon test (p values for each group are listed below: T cell: p=0.661; epithelial cell: p=0.039; neutrophil: p=0.026; macrophage: p=0.661; fibroblast: p=0.661; plasma: p=0.078; endothelial cell: p=0.851; B cell: p=0.104; mast cell: p=0.753). Statistics are shown as *p<0.05; **p<0.01; n.s., not significant.

The online version of this article includes the following source data and figure supplement(s) for figure 1:

**Source data 1.** Annotated code for data used in *Figure 1*.

**Figure supplement 1.** The single-cell genomic atlas of cervical cancer.

*1B*). In summary, these data describes primarily a complex microenvironment in CC, especially in the ADC type, with varied composition of immune and tumor cells under different conditions.

## Sub-clusters of epithelial cells in ADC exhibit elevated stem-like features

The aforementioned data has shown that the majority of the CC tumor is composed of epithelial cells. In order to explore the characteristics of these epithelial cells within the TIME of ADC, we further classified them into 12 sub-clusters based on the subsets of differently expressed genes (DEGs) in each group (*Figure 2A*). Each sub-cluster was named based on the most prominently upregulated gene in each panel of signature genes (*Figure 2B and C*). Several epithelial cell clusters were strongly enriched in ADC than SCC, such as Epi_04_TFF2, Epi_06_TMC5, Epi_07_CAPS, and Epi_08_SCGB3A1. Notably, Epi_09_SST, Epi_10_CYSTM1, Epi_11_REG1A, and Epi_12_RRAD seemed to be exclusively enriched in ADC, not in SCC (*Figure 2D*). The stratified enrichment of different cluster between ADC and SCC has captured our attention for further investigation. Additionally, Epi_09_SST, Epi_10_CYSTM1, and Epi_11_REG1A were solely enriched in HPV-positive cases, although no distinctive clusters had been found in HPV-negative patients (*Figure 2—figure supplement 1A*). When comparing samples from patients in the early stages with those who have LN metastasis, three clusters (Epi_02_IGLC2, Epi_05_CCL5, and Epi_12_RRAD) were slightly increased among the late-stage patients. Surprisingly, we found that the sub-cluster Epi_10_CYSTM1 was exclusively present in stage IIIC patients, indicating that it might be a potential target for identification of the biomarkers for late-stage patients (*Figure 2—figure supplement 1B*).

We further evaluated the features of each epithelial cluster, especially their predicted roles in regulating tumorigenesis. To assess the differentiation potential of each cell cluster, we employed the Cellular Trajectory Reconstruction Analysis using gene Counts and Expression (CytoTRACE) method (*Figure 2E*). The results showed that clusters of Epi_07_CAPS, Epi_09_SST, Epi_10_CYSTM1, and Epi_12_RRAD ranked the top 4 in CytoTRACE score, indicating the potential for pluripotency and differentiation, also referred to stem-like features. Interestingly, Epi_09_SST, Epi_10_CYSTM1, and Epi_12_RRAD were exclusively enriched in ADC. The proportion of cluster Epi_07_CAPS, which acquired the highest level of stemness, was higher in ADC, although Epi_07_CAPS was also identified in SCC (*Figure 2D*). Subsequently, pseudotime analysis was conducted to elucidate the differentiation trajectories. The results showed that Epi_07_CAPS, Epi_09_SST, and Epi_10_CYSTM1 were positioned toward the end of pseudotime developmental trajectory, which was in consistence with their high stem-like characteristics and indicated a higher degree of malignancy in ADC than SCC (*Figure 2—figure supplement 1C*). In addition, we performed the malignancy scoring analysis, which demonstrated that cluster Epi_10_CYSTM1 was predicted to exhibit the highest degree of malignancy compared to other clusters (*Figure 2F*). Interestingly, cluster Epi_10_CYSTM1 displayed a diverse developmental profile and was exclusively identified in ADC. Conversely, Epi_07_CAPS and Epi_12_RRAD, which exhibited high stem-like characteristics, were ranking lower in terms of malignancy scoring compared to other clusters. The observation of cluster Epi_10_CYSTM1 and its potential specificity to ADC raises the question of whether this cluster is associated with the aggressiveness of ADC.

Besides, we performed Gene Ontology Biological Process (GOBP) analysis on specific signaling pathways based on the gene expression pattern of cluster Epi_10_CYSTM1 (*Figure 2G*). The results

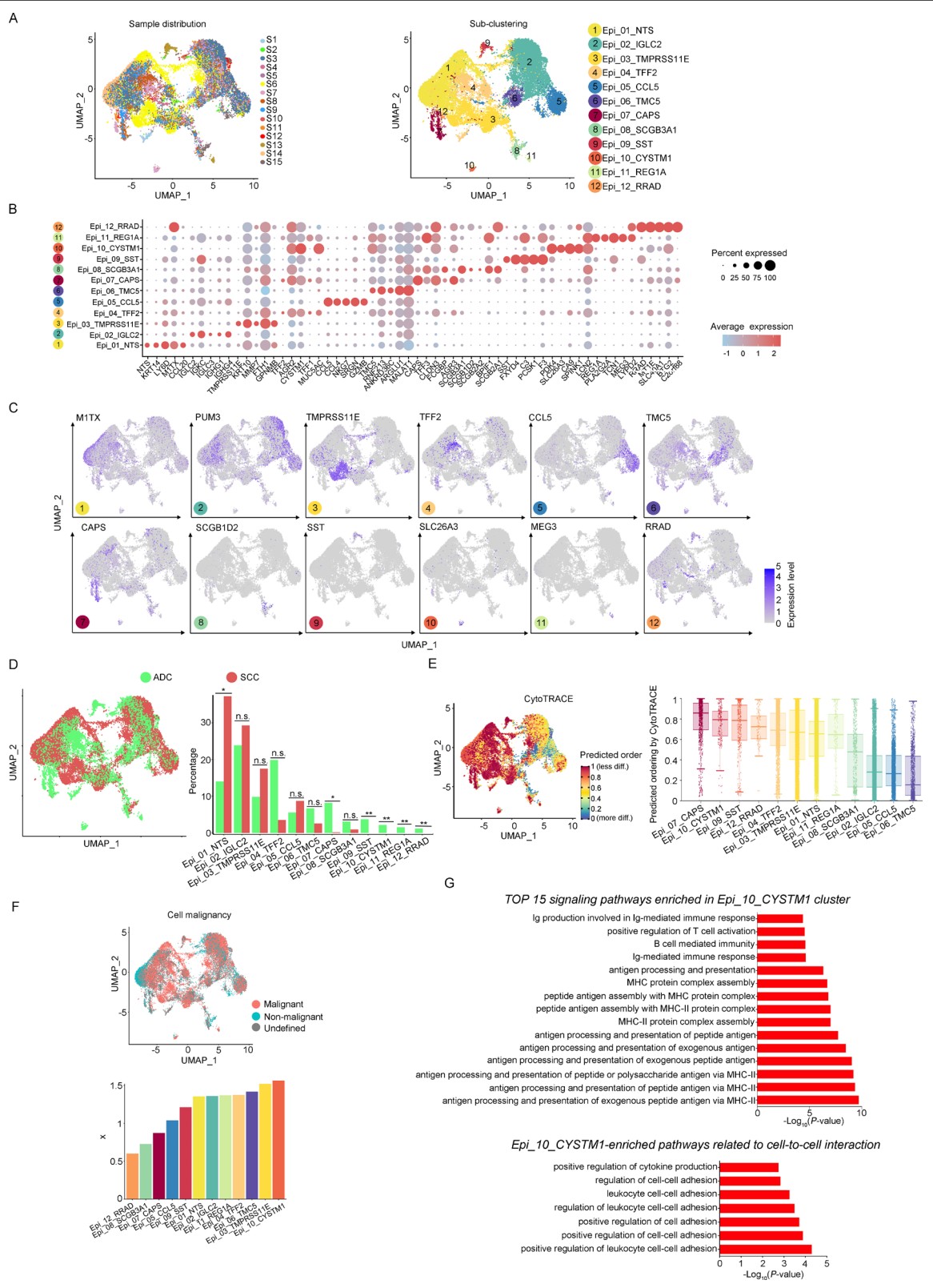

**Figure 2.** The scRNA-seq data reveals the malignant features of tumor epithelial cells. (**A**) Uniform Manifold Approximation and Projection (UMAP) plots showing the distribution of epithelial cells in 15 samples and the sub-clustering into 12 clusters according to ectopic gene expressions. Each cluster of epithelial cells was named using the most highly enriched gene. (**B**) Heatmap plot showing the annotation of each epithelial sub-cluster with top 5 differently expressed genes (DEGs). (**C**) UMAP plots demonstrating the most specific gene as the marker for each sub-cluster. (**D**) UMAP (left) and

*Figure 2 continued on next page*

*Figure 2 continued*

histogram (right) plots to compare the differences of distribution and proportion of each epithelial cell sub-cluster between different histological types (adenocarcinoma [ADC] vs. squamous cell carcinoma [SCC]). Statistics were performed using R software with two-sided Wilcoxon test (p values for each group are listed below: Epi_01_NTS: p=0.040; Epi_02_IGLC2: p=0.661; Epi_03_TMRPSS11E: p=0.661; Epi_04_TTF2: p=0.104; Epi_05_CCL5: p=0.950; Epi_06_TMC5: p=0.412; Epi_07_CAPS: p=0.040; Epi_08_SCGB3A1: p=0.412; Epi_09_SST: p<0.001; Epi_10_CYSTM1: p<0.001; Epi_11_REG1A: p<0.001; Epi_12_RRAD: p<0.001). Statistics were shown as *p<0.05; **p<0.01; n.s., not significant. (**E**) UMAP plot of CytoTRACE showing the thermal imaging projection of predicted developmental order (left) and histogram plot showing the ranking of CytoTRACE scores (right). (**F**) UMAP plot of malignancy analysis demonstrating the cell malignancy features (up) and ranking of scores (down). (**G**) GOBP analyses showing the top 15 enriched signaling pathways of Epi_10_CYSTM1 that are more active in ADC than SCC (up), particularly pathways related to cell-to-cell interactions (down). The histogram data are transformed from -Log$_{10}$(p-value) for visualization. Ig: immunoglobulin; MHC: major histocompatibility complex.

The online version of this article includes the following source data and figure supplement(s) for figure 2:

**Source data 1.** Annotated code for data used in *Figure 2*.

**Figure supplement 1.** The scRNA-seq data reveals the malignant features of tumor epithelial cells.

revealed that cluster Epi_10_CYSTM1 tended to be more active in regulating immunity-related pathways, which indicated this cluster might be possibly associated with dysfunction of immunity. In addition, cluster Epi_10_CYSTM1 was predicted to be more active in regulating cell-to-cell interaction (*Figure 2G*). Therefore, further investigation is warranted to explore the intricate crosstalk networks between epithelial cells and other cell types in the TIME of ADC.

## ADC-specific epithelial cluster-derived gene SLC26A3 is a potential prognostic marker for lymph node metastasis

Based on the aggressive characteristics of cluster Epi_10_CYSTM1 from above bioinformatic predictions, we further examined the correlation between this cluster and the clinical features of CC. For validation, we alternatively examined cluster Epi_12_RRAD (*Figure 3—figure supplement 1A*), which was another malignant cluster predominantly found in ADC (*Figure 2D*). We firstly checked the DEGs of these two clusters. In Epi_12_RRAD cluster, we identified Insulin-like growth factor 2 (IGF2) and alcohol dehydrogenase 1C (ADH1C), which were exclusively expressed in Epi_12_RRAD, as two most specific genes of this cluster (*Figure 3—figure supplement 1B*). However, the immunohistochemistry (IHC) staining showed that IGF2 was almost negative across cases with different clinical stages (*Figure 3—figure supplement 1C*). On the other hand, ADH1C was strongly expressed in both early-stage and late-stage cases (*Figure 3—figure supplement 1C*). Likewise, the survival analysis of Epi_12_RRAD-enriched gene signatures did not demonstrate a significant difference between the two groups (*Figure 3—figure supplement 1D*). These results suggest that cluster Epi_12_RRAD may not be a potential target when relating to the clinical features of ADC.

In the cluster of Epi_10_CYSTM1, the DEG with the highest expression is CYSTM1 (*Figure 3A*). However, CYSTM1 was also expressed in other cell clusters and showed low specificity (*Figure 3B*). On the other hand, SLC26A3, ORM1, and ORM2 were specifically enriched in Epi_10_CYSTM1 and could be considered as potential candidate biomarkers for representation of this cluster. Nevertheless, ORM1/OMR2 were positively expressed but were not satisfied to distinguish the severity of CC cases due to an irrelevance between expression intensity and clinical stages (*Figure 3—figure supplement 1E*). Interestingly, SLC26A3, as a representative marker of cluster Epi_10_CYSTM1, showed potential value to associated with late clinical stages for CC patients (*Figure 3—figure supplement 1E*). This is encouraging since currently it lacks biomarkers to predict late stages of CC and the only way to detect LN metastasis beforehand is using radiological tools, which have technical limitations.

This is a problem because misdiagnosis of clinical staging affects the disease outcome and treatment approach for CC patients. In order to examine the fact of post-surgical upstaging, we merged data from two independent patient cohorts obtained from two clinical centers: Xiangya Hospital (*Cohort 1*) and Hunan Cancer Hospital (*Cohort 2*). The criterion for objective selection is outlined in *Figure 3—figure supplement 1F*. The rates of upstaging for each center are 10.85% (23 out of 212 in *Cohort 1*) and 15.31% (214 out of 1398 in *Cohort 2*), respectively. These rates fall within the range reported by previous studies (*Dabi et al., 2018*; *Thelissen et al., 2022*). Interestingly, both cohorts of data revealed that the misdiagnose rate was significantly higher in HPV-negative patients than HPV-positive patients, especially among ADC patients. From the data of *Cohort 2*, it was observed that ADC patients were more likely to be misdiagnosed, which means that metastatic LNs were more

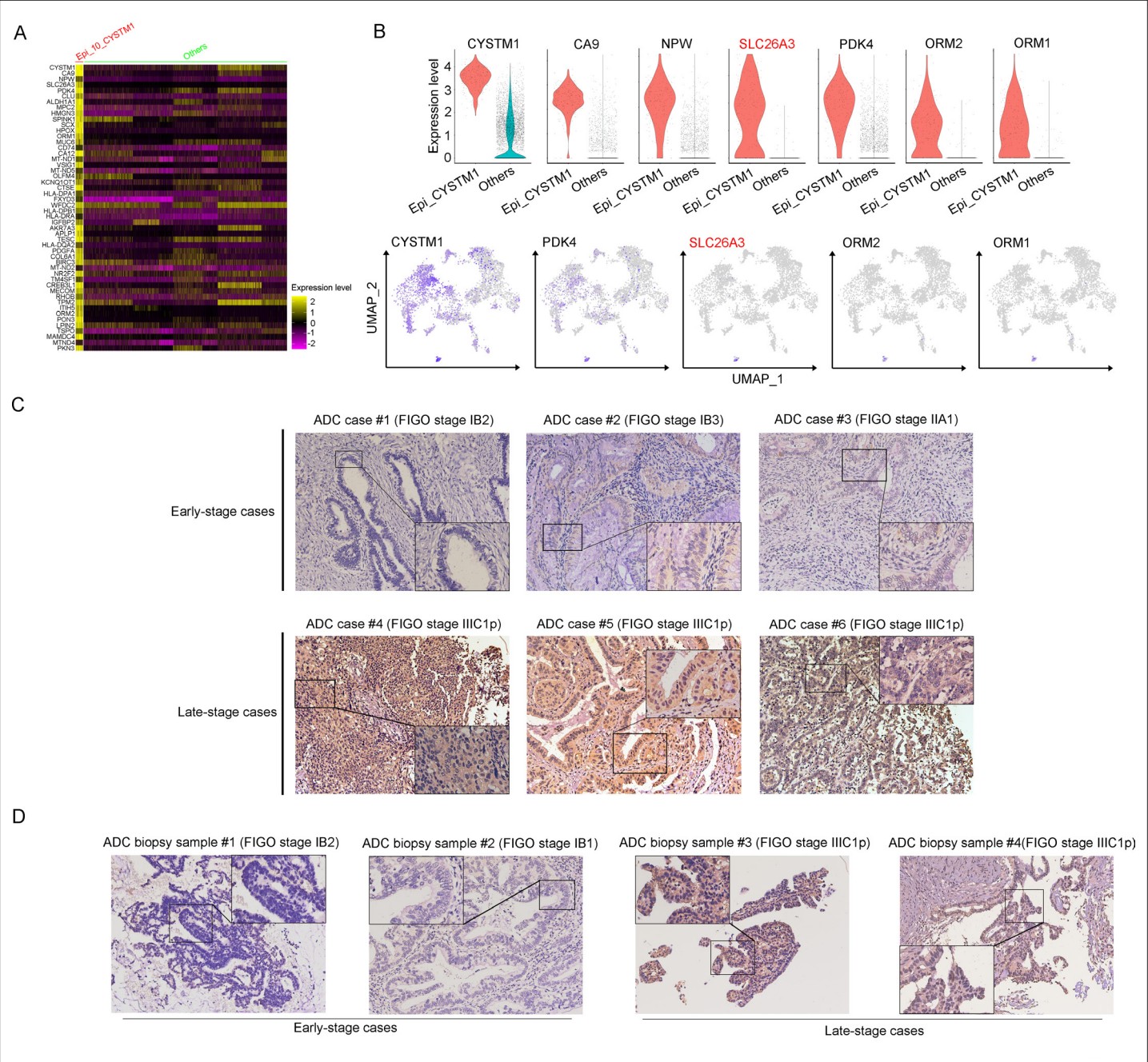

**Figure 3.** SLC26A3 is identified as a potential prognostic and diagnostic indicator for lymph node metastasis of cervical cancer (CC) patients. (**A**) Heatmap showing the top 50 differently expressed genes (DEGs) of Epi_10_CYSTM1 cluster in comparison to all the other sub-clusters (red: Epi_10_CYSTM1 cluster; green: other epithelial cell sub-clusters). (**B**) Violin plots showing the expression difference between two groups (top), with Uniform Manifold Approximation and Projection (UMAP) plots (bottom) demonstrating the specificity of each candidate gene, filtering SLC26A3 as the most identical marker for this sub-cluster. (**C**) Immunohistochemistry (IHC) staining showing the protein expression of SLC26A3 in surgically resected adenocarcinoma (ADC) samples which are classified as early stages (International Federation of Gynecology and Obstetrics [FIGO] stage I–IIA) and late stages (FIGO stage IIIC1–2p). Images from six individual cases are shown as representatives for each group. (**D**) IHC staining showing the protein expression of SLC26A3 in biopsy ADC samples which are classified as early stages (FIGO stage I–IIA) and late stages (FIGO stage IIIC1–2p). Images from four individual cases are shown as representatives for each group. The method of *H*-score is used and the scoring system is as follows: negative (0), weak (1), intermediate (2), and strong (3). Expression is quantified by the *H*-score method.

The online version of this article includes the following figure supplement(s) for figure 3:

*Figure 3 continued on next page*

*Figure 3 continued*

**Figure supplement 1.** SLC26A3 is identified as a potential prognostic and diagnostic indicator for lymph node metastasis of cervical cancer (CC) patients.

**Figure supplement 2.** Full-scan image of the slide from adenocarcinoma (ADC) case #5 in *Figure 3C*.

difficult to detect in ADC patients via radiological tools. In *Cohort 1*, we have also observed the same tendency although without significant difference between different histological types, probably due to a smaller sample size (*Tables 1 and 2*).

As one of stage IIIC-specific (*Figure 2—figure supplement 1B*) cell clusters, Epi_10_CYSTM1, with its representative marker gene SLC26A3, presents potential diagnostic value to predict LN metastasis, which has been shown above (*Figure 3B*, *Figure 3—figure supplement 1E*). Therefore, we try to validate SLC26A3's association with staging of IIIC, via detecting the expression pattern on post-surgical ADC samples, and more practically on biopsy samples. The results from IHC staining showed that ADC patients with stage IIIC had a higher expression of SLC26A3 (*Figure 3C and D*, *Tables 3 and 4*). In summary, our results propose that SLC26A3 might be considered a diagnostic marker to predict LN metastasis in ADC patients.

## Enrichment of regulatory T cells indicates an immunosuppressive status of ADC

T cells play a crucial role as the primary effectors of tumor immunity. Our data reveals that T cells constitute the predominant cell population within the TIME of CC (*Figure 1F*). A total of 52,297 T cells were categorized into eight main clusters (*Figure 4A*). The gene expression patterns of T cells in CC were in accord with those observed in other types of cancers (*Sorin et al., 2023*; *Xue et al., 2022*). Based on canonical genes (*Li et al., 2022*), these cells were re-clustered as the following four types of T cells: exhausted T cells (marked with CD8, TIGIT, PDCD1, LAG3, HAVCR2), cytotoxic T cells (marked with CD8 and CD3), regulatory T cells (Treg, marked with CD4, FOXP3, and IL2RA),

**Table 1.** The association between post-surgical upstaging and clinical characteristics in patient *Cohort 1* (from Xiangya Hospital).

| Characteristics | Case number | Upstaging after surgery | | $\chi^2$ | p value |
| | | Yes | No | | |
| --- | --- | --- | --- | --- | --- |
| *Total* | 215 | | | | |
| *Histological types* | 209 | | | | |
| ADC | 55 | 8 (14.55%) | 47 (85.45%) | 0.955 | 0.328 |
| SCC | 154 | 15 (9.74%) | 139 (90.26%) | | |
| *HPV status* | 196 | | | | |
| HPV+ | 181 | 18 (9.94%) | 163 (90.06%) | 3.887 | 0.049* |
| HPV- | 15 | 4 (26.67%) | 11 (73.33%) | | |
| *ADC with HPV status* | 50 | | | | |
| HPV+ | 40 | 4 (10.00%) | 36 (90.00%) | 5.357 | 0.021* |
| HPV- | 10 | 4 (40.00%) | 6 (60.00%) | | |
| *SCC with HPV status* | 143 | | | | |
| HPV+ | 138 | 13 (9.42%) | 125 (90.58%) | 0.518 | 0.472 |
| HPV- | 5 | 0 (0.00%) | 5 (100%) | | |

In each comparison panel, patients with insufficient data were excluded from analysis. The total number is 215, among which 6 cases are specific histological types (3 are neuroendocrine carcinoma [Neuro]; 2 are adenosquamous carcinoma; 1 is malignant melanoma) and has been excluded for comparison between ADC and SCC. Because the HPV status information is unclear in 16 cases, a total number of 196 cases have been used for comparison between different HPV statuses. As for sub-classification between different HPV status under different histological types, three Neuro cases have been excluded.

*p<0.05 is considered significantly different. p<0.01 is presented as **.

ADC, adenocarcinoma; HPV, human papillomavirus; SCC, squamous cell carcinoma.

**Table 2.** The association between post-surgical upstaging and clinical characteristics in patient *Cohort 2* (from Hunan Cancer Hospital).

| Characteristics | Case number | Upstaging after surgery | | $\chi^2$ | p value |
| | | Yes | No | | |
| --- | --- | --- | --- | --- | --- |
| *Total* | 1398 | | | | |
| *Histological types* | 1336 | | | | |
| ADC | 190 | 40 (21.05%) | 150 (78.95%) | 4.911 | 0.027* |
| SCC | 1146 | 169 (14.75%) | 977 (85.25%) | | |
| *HPV status* | 1166 | | | | |
| HPV+ | 1031 | 131 (12.71%) | 900 (87.29%) | 66.616 | <0.001** |
| HPV- | 135 | 54 (40.00%) | 81 (60.00%) | | |
| *ADC with HPV status* | 162 | | | | |
| HPV+ | 131 | 18 (13.74%) | 113 (86.26%) | 24.999 | <0.001** |
| HPV- | 31 | 17 (54.84%) | 14 (45.16%) | | |
| *SCC with HPV status* | 958 | | | | |
| HPV+ | 864 | 109 (12.62%) | 755 (87.38%) | 40.229 | <0.001** |
| HPV- | 94 | 35 (37.23%) | 59 (62.77%) | | |
| *Neuro with HPV status* | 27 | | | | |
| HPV+ | 21 | 3 (14.29%) | 18 (85.71%) | 0.964 | 0.326 |
| HPV- | 6 | 0 (0.00%) | 6 (100.00%) | | |
| *ADSCC with HPV status* | 14 | | | | |
| HPV+ | 12 | 1 (8.33%) | 11 (91.67%) | 2.431 | 0.119 |
| HPV- | 2 | 1 (50.00%) | 1 (50.00%) | | |

In each comparison panel, patients with insufficient data were excluded from analysis. The total number is 1398 and 62 cases have been excluded for comparison between ADC and SCC, due to special pathological types (carcinosarcoma: 1; melanoma: 1; neuroendocrine carcinoma: 38; basaloma: 1; adenosquamous carcinoma: 18; malignant peripheral nerve sheath tumor: 1; fibroblastic sarcoma: 1; adenosarcoma: 1). Because the HPV status information is unclear in 232 cases, a total number of 1166 cases has been used for comparison between different HPV status. As for subclassification between two HPV statuses under different histological types, the excluded case numbers for each group are as follows due to unknown HPV status: 28 in ADC; 188 in SCC; 11 in Neuro; 4 in ADSCC (*Figure 3—figure supplement 1*).

*p<0.05 is considered significantly different. p<0.01 is presented as **.

ADC, adenocarcinoma; ADSCC, adenosquamous carcinoma of cervix; HPV, human papillomavirus; Neuro, neuroendocrine carcinoma; SCC, squamous cell carcinoma.

and activated T cells (marked with CD8, C69, and CD3G) (*Figure 4A–C*). The subsets of DEGs from each type of T cells are shown in *Figure 4—figure supplement 1C*. When comparing the checkpoint pathway states among different T cell clusters, we observed relatively higher levels of LAG3 and PDCD1 (PD1) in exhausted T cells, while CTLA4 and TIG1 were more highly expressed in Tregs. These genes were identified to present immunosuppressive functions (*Qiu et al., 2023*; *Figure 4D*). On the other hand, genes such as ICOS and TNFRSF, which indicate the activation of immune checkpoint pathways, exhibited low expression levels in cytotoxic T cells and activated T cells (*Figure 4D*). These findings lead us to hypothesize that the establishment of an immunosuppressive TIME in ADC may involve the recruitment of regulatory T cells (Tregs) to the tumor area and the subsequent inactivation of a substantial proportion of cytotoxic T cells.

We further compared different status of CC with different attribution of T cell subtypes. It was demonstrated that ADC had a significantly higher proportion of Tregs compared to SCC (*Figure 4E*). Meanwhile, ADC patients who tested negative for HPV showed a higher enrichment of Tregs and a lower proportion of activated T cells compared to those with HPV-positive status (*Figure 4—figure*

**Table 3.** The association between clinical characteristics and SLC26A3 protein expression via IHC tested on post-surgical samples.

| Characteristics | Case number | SLC26A3 expression | | $\chi^2$ | p value |
|---|---|---|---|---|---|
| | | High | Low | | |
| *Total* | 56 | | | | |
| *Age* | 56 | | | | |
| ≥50 | 28 | 8 (28.57%) | 20 (71.43%) | 0.717 | 0.397 |
| <50 | 28 | 11 (39.29%) | 17 (60.71%) | | |
| *Histological grading* | 51 | | | | |
| G1, G1-2 | 20 | 5 (25.00%) | 15 (75.00%) | 1.527 | 0.217 |
| G2, G2-3, G3 | 31 | 13 (41.94%) | 18 (50.06%) | | |
| *FIGO stages* | 56 | | | | |
| I–II (LN-M-) | 47 | 12 (25.53%) | 35 (74.47%) | 9.198 | 0.002** |
| IIICp (LN-M+) | 9 | 7 (77.78%) | 2 (22.22%) | | |
| *HPV status* | 53 | | | | |
| HPV+ | 44 | 13 (29.55%) | 31 (70.45%) | 2.254 | 0.133 |
| HPV- | 9 | 5 (55.56%) | 4 (44.44%) | | |

*p<0.05 is considered significantly different. p<0.01 is presented as **.

FIGO, International Federation of Gynecology and Obstetrics; HPV, human papillomavirus; IHC, immunohistochemistry.

supplement 1). These findings shed light on the underlying mechanism for immunotherapy insensitivity of ADC, especially in HPV-negative status.

## Tumor-associated neutrophils (TANs) surrounding ADC tumor area may contribute to the formation of a malignant microenvironment

The large-scale enrichment of specific gene subsets in neutrophils in CC prompted us to further examine their roles. A total of 12,586 neutrophils were detected and the composition proportion in ADC (20.05%) was almost five times higher than that in SCC (4.23%) (*Figure 1F*). Based on this observation, we hypothesize that ADC-specific neutrophil clusters may be related to poor prognosis and increased malignancy. All the cells were initially grouped into six clusters based on the gene expression modules (*Figure 5A*). In the context of cancer, TANs can exhibit dual functions, either promoting tumor (pro-tumor TANs) or inhibiting tumor progression (anti-tumor TANs). Based on the consensus markers identified by *Jaillon et al., 2020* and *Xue et al., 2022*, we performed further clustering of the neutrophils in our dataset, resulting in the identification of three sub-types of TANs: pro-tumor TANs (Neu_03, marked with CD66, CD11, and MT1X), anti-tumor TANs (Neu_02, marked with CD66, CD11, CCL5, KRT5, and ELF3), TANs with isg (interferon-stimulated genes), and other undefined types (Neu_01, distinguished by CD66, CD11, IFITM2, S100A8, and LST1) (*Figure 5B and C*). Generally, the Neu_01 sub-cluster, which consists of TANs with isg with the subset of specific genes, accounted for the largest proportion among all neutrophils. Notably, the presence of pro-tumor TANs was significantly higher in ADC compared to SCC (*Figure 5D*), as well as in HPV-negative cases compared to HPV-positive cases (*Figure 5—figure supplement 1A*). On the other hand, the proportion of anti-tumor TANs in ADC was predominantly lower than that in SCC (*Figure 5D*).

We further conducted the GOBP analysis on gene subsets derived from pro-tumor TANs to predict their potential functions. The findings indicated that pro-tumor TANs in ADC might be more active in regulating cell-to-cell interactions and chemotaxis. Specifically, they were involved in pathways such as chemokine-mediated or cytokine-mediated signaling, neutrophil activation, and receptor internalization (*Figure 5E*). These results suggest that pro-tumor TANs may engage in communication with various cell types. Interestingly, pro-tumor TANs were predicted to acquire enhanced activation in mediating inhibitory pathways related to immunity (*Figure 5E*). These pathways include the negative regulation of leukocyte proliferation and lymphocyte activation, as well as the suppression of IL-10

**Table 4.** The association between clinical characteristics and SLC26A3 protein expression via IHC tested on biopsy small specimens.

| Characteristics | Case number | SLC26A3 expression | | $\chi^2$ | p value |
|---|---|---|---|---|---|
| | | High | Low | | |
| Total | 43 | | | | |
| *Age* | | | | | |
| ≥50 | 16 | 3 (18.75%) | 13 (81.25%) | 4.605 | 0.032 |
| <50 | 27 | 14 (51.85%) | 13 (48.15%) | | |
| *Histological grading* | | | | | |
| G1, G1–2 | 9 | 4 (44.44%) | 5 (55.56%) | 0.115 | 0.735 |
| G2, G2–3, G3 | 34 | 13 (38.24%) | 21 (61.76%) | | |
| *FIGO stages* | | | | | |
| I–II (consistent staging) | 29 | 7 (24.14%) | 22 (75.86%) | 8.833 | 0.003* |
| IIICp (upstaging) | 14 | 10 (71.43%) | 4 (28.57%) | | |
| *HPV status* | | | | | |
| HPV+ | 32 | 10 (31.25%) | 22 (68.75%) | 3.592 | 0.058 |
| HPV- | 11 | 7 (63.64%) | 4 (36.36%) | | |

*p<0.05 is considered significantly different. p<0.01 is presented as **.
FIGO, International Federation of Gynecology and Obstetrics; HPV, human papillomavirus; IHC, immunohistochemistry.

production, which is known to promote immune responses (*Saraiva et al., 2020*; *Sarvaria et al., 2017*).

The role of pro-tumor TANs in association with disease prognosis was further investigated using the TCGA database. The patients with higher expression of gene panels specific to pro-tumor TANs exhibited a poorer prognosis compared to those with lower expression (*Figure 5F*). These findings suggest that the increased malignancy observed in ADC, particularly in HPV-negative cases, may also be related to the enrichment of pro-tumor TANs.

## Cellular heterogeneity of plasma/B cells in ADC

In our study, we identified six distinctive clusters of plasma/B cells based on the set of gene enrichment: Plasma/B_01 (marked with IGHA2, IGHG2, and IGLC2), Plasma/B_02 (marked with HLA-DRA, HLA-DPB, and CD37), Plasma/B_03 (marked with KRT17 and S100A2), Plasma/B_04 (marked with CCL5, CXCL13, IL-32, and CCL4), Plasma/B_05 (marked with CXCL8), and Plasma/B_06 (marked with HMGB2) (*Figure 6A–C*). Among these clusters, the most abundant one was Plasma/B_01_IGHA2 (*Figure 6D*, *Figure 6—figure supplement 1A and B*). Interestingly, the proportion of Plasma/B_01_IGHA2 was found to be higher in ADC compared to SCC, and it was also slightly higher in HPV-negative cases than in HPV-positive cases (*Figure 6D*, *Figure 6—figure supplement 1A*). Generally, B-lymphocytes are known to have tumor-inhibitory properties. However, the ectopic gene expression module of plasma/B cells in ADC suggested that some sub-clusters might present a tumor-promoting role. When we combined the top 50 signature genes in Plasma/B_01_IGHA2-enriched subset and performed the survival analysis using TCGA database, we observed that higher expression of the signature genes predicted a poorer prognosis of CC patients (*Figure 6E*). Taken together, the cellular heterogeneity of plasma/B cells in the TIME of ADC is identified at the single-cell level.

## Crosstalk among tumor cells, Tregs, and neutrophils establishes the immunosuppressive TIME in ADC

In the above description, we have outlined that certain ADC-enriched cell sub-clusters, such as Epi_10_CYSTM1, Tregs, and pro-tumor TANs, might play oncogenic roles in regulating various cellular activities, including maintenance of stemness and evasion of immune surveillance. To gain deeper insights into the interplay among these clusters and their influence on ADC progression, we harnessed

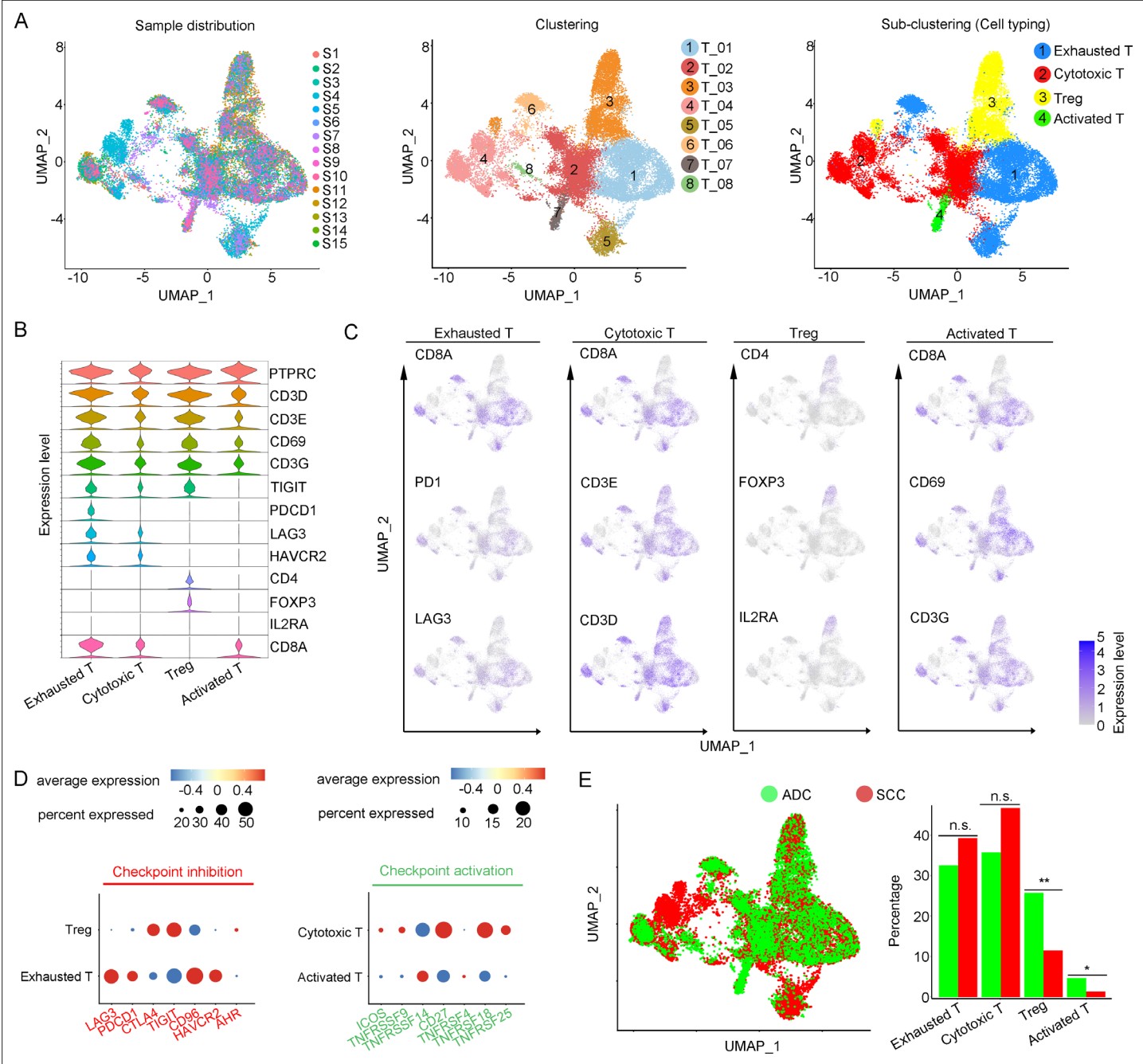

**Figure 4.** Cellular and molecular heterogeneity of T cells in adenocarcinoma (ADC). (**A**) Uniform Manifold Approximation and Projection (UMAP) plots showing the sample distribution (left), original cell clustering of T cells (middle, eight clusters in total) and re-clustering of T cell subtypes (right, four major sub-clusters: exhausted T cell, cytotoxic T cell, regulatory T cell, and activated T cell) according to acknowledged marker genes. (**B**) Violin plot showing the expression of specific marker genes that annotate each sub-type of T cells. (**C**) UMAP plots showing the widely recognized classification markers that denote each type of T cell sub-cluster. (**D**) Dot heatmap plots that demonstrate the level of marker genes representing the signaling pathways of immune checkpoint activation and inhibition for grouped sub-clusters. (**E**) Differences of distribution and proportion of each T cell sub-cluster between different histological types (ADC vs. squamous cell carcinoma [SCC]). Statistics were performed using R software with two-sided Wilcoxon test (p values for each group are listed below: exhausted T: p=0.571; cytotoxic T: p=0.078; Treg: p<0.01; activated T: p=0.040). Statistics are shown as *p<0.05; **p<0.01; n.s., not significant. Treg: regulatory T cell.

The online version of this article includes the following source data and figure supplement(s) for figure 4:

**Source data 1.** Annotated code for data used in *Figure 4*.

**Figure supplement 1.** Cellular and molecular heterogeneity of T cells in adenocarcinoma (ADC).

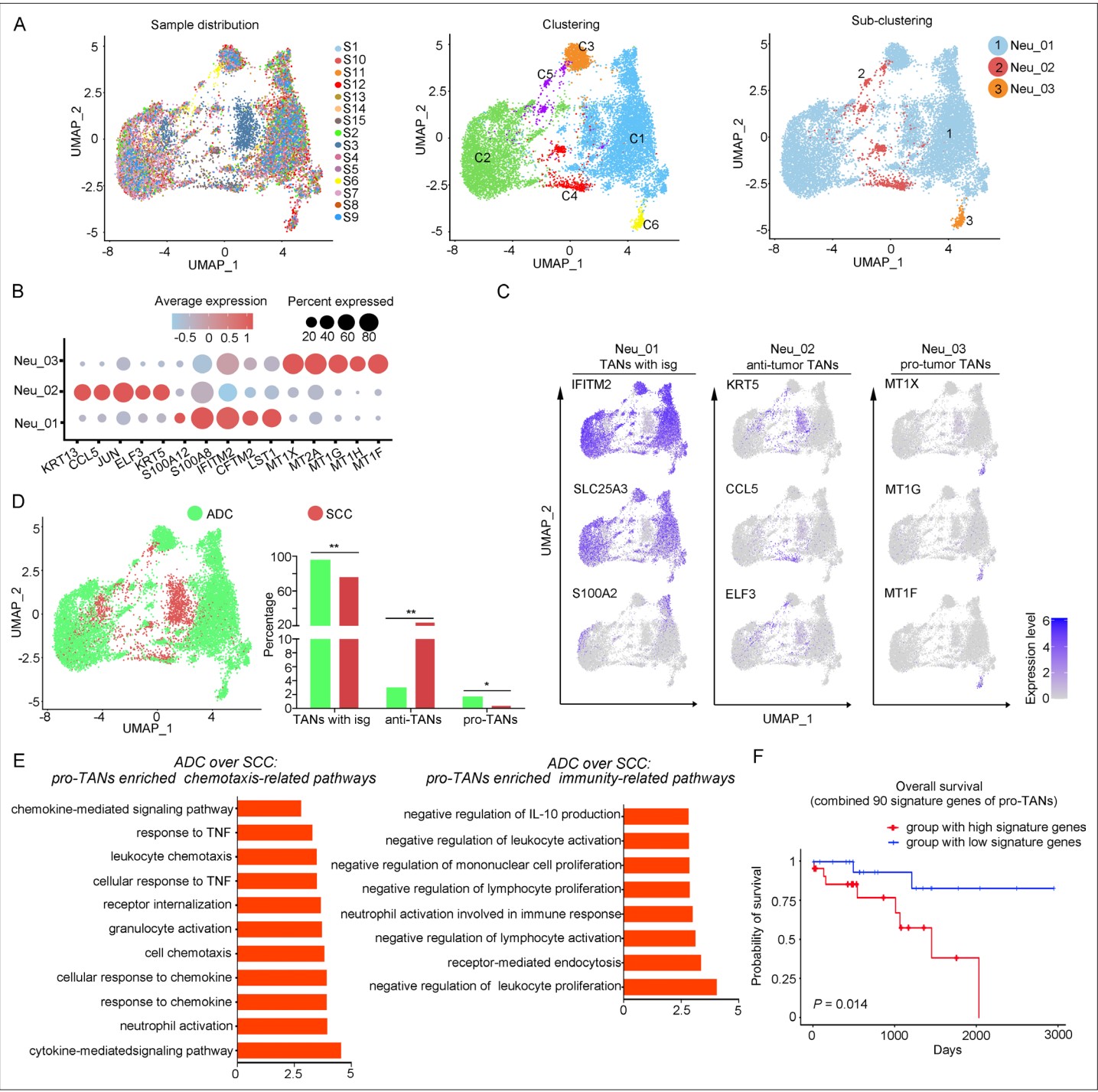

**Figure 5.** The heterogeneity of tumor-associated neutrophils (TANs) in adenocarcinoma (ADC). (**A**) Uniform Manifold Approximation and Projection (UMAP) plots showing the distribution of neutrophils in 15 samples (left), original clustering (middle) the re-clustering (right) into three sub-clusters according to gene markers of TANs. (**B**) Dot heatmap showing the top 5 differently expressed genes (DEGs) in each sub-cluster of TANs. (**C**) UMAP plots for annotation of each sub-type of TANs with published marker genes presented: pro-tumor TANs, anti-tumor TANs, and TANs with isg. (**D**) UMAP and histogram plots to compare the differences of distribution and proportion of each sub-types of TANs between different histological types (ADC vs. squamous cell carcinoma [SCC]). Statistics were performed using R software with two-sided Wilcoxon test (p values for each group are listed below: TANs with isg: p=0.002; anti-TANs: p=0.002; pro-TANs: p=0.049). Statistics are shown as *p<0.05; **p<0.01; n.s., not significant. (**E**) Gene Ontology Biological Process (GOBP) analyses of signaling pathways that are more active in ADC than SCC, in terms of the pro-tumor TANs cluster. The histogram data are transformed from -Log$_{10}$ (p-value). (**F**) Kaplan–Meier curve showing the overall survival rate of CC patients stratified by the top 90 genes-scaled signature of pro-TANs.

*Figure 5 continued on next page*

*Figure 5 continued*

The online version of this article includes the following source data and figure supplement(s) for figure 5:

**Source data 1.** Annotated code for data used in *Figure 5*.

**Figure supplement 1.** The heterogeneity of tumor-associated neutrophils (TANs) in adenocarcinoma (ADC).

the CellChat tool (*Jin et al., 2021*). Our primary focus revolved around unraveling the mechanisms governing Treg recruitment to the tumor microenvironment and the maintenance of stemness within ADC-enriched cell sub-clusters.

Given that the three ADC-specific sub-clusters (Epi_07_CAPS, Epi_10_CYSTM1, and Epi_12_RRAD) of epithelial cells were predicted to exhibit high levels of stem-like properties, and particularly that Epi_10_CYSTM1 presented the highest degree of malignancy among them, we further analyzed their possible interactions with other types of cells. Firstly, in comparison to SCC, ADC-enriched epithelial sub-clusters exhibited a higher propensity for interaction with Tregs through ligand-receptor signaling pathways, including ALCAM (ALCAM>>CD6) and MHC-II (HLA-DR>>CD4) (*Figure 7A–E*), which have been reported to be essential for Treg recruitment, expansion, and stabilization to establish the immunosuppressive microenvironment (*Chalmers et al., 2022*; *Ferragut et al., 2021*; *Freitas et al., 2019*). To further uncover the underlying reason for the enhanced stemness observed in ADC epithelial cells, we focused on the signaling received by Epi_10_CYSTM1 cell cluster. Interestingly, the CellChat analysis revealed that, compared to other pathways, such as CD46>>JAG1 and GZMA>>PARD3, the interaction of TGFβ>>TGFBR between Tregs and epithelial cells (particularly in cluster Epi_10_CYSTM1) was activated in ADC but not in SCC (*Figure 7F and G*). TGFβ is a canonical secreted protein that induces carcinogenesis, such as cancer cell proliferation, invasion, self-renewal, and epithelial-to-mesenchymal transition (EMT) (*Derynck et al., 2021*; *Massagué, 2008*; *Tauriello et al., 2022*). Thus, we speculate that the recruited Tregs might secrete TGFβ to stimulate the tumor epithelial cells of ADC, leading to increased stemness. Additionally, pro-tumor TANs were also recruited to the tumor area and interacted with Epi_10_CYSTM1, as well as Epi_07_CAPS and Epi_12_RRAD cells, via the ANNEXIN-to-FPR1/FPR2 signaling pathway (*Figure 7—figure supplement 1A and B*). Previous studies have reported that the ANNEXIN pathway can promote the chemotaxis of neutrophils (*Araújo et al., 2021*). Likewise, TANs expressing isg, which have been shown to promote tumorigenesis as mentioned earlier (*Figure 5—figure supplement 1C*), were also recruited to the tumor area through the ANNEXIN pathway. These findings suggest that the aggressive clusters of TANs may contribute to the formation of an ADC-specific TIME.

To test these hypotheses, we conducted immunofluorescence (IF) analysis and observed a greater recruitment of Tregs (marked by FOXP3) in close proximity to regions exhibiting high SLC26A3 expression (Epi_10_CYSTM1 cell cluster) (*Figure 7H*). Furthermore, when compared to regions with low SLC26A3 expression, tumors displaying high SLC26A3 levels exhibited significantly increased expression of Aldehyde Dehydrogenase 1 Family Member A1 (ALDH1A1), a known marker for cancer stem cells in CC (*Douville et al., 2009*; *Figure 7—figure supplement 1C*). Notably, we identified SLC26A3[high] tumor cells expressing ALCAM in close proximity to CD6+Tregs, suggesting potential involvement of ALCAM>>CD6 signaling in Treg recruitment (*Figure 7I*). Additionally, we observed reduced E-cadherin expression in tumor cells within SLC26A3[high] regions, possibly induced by TGFβ secreted by adjacent Tregs (*Figure 7J*, *Figure 7—figure supplement 1D*). Altogether, these findings confirm the intricate crosstalk between the Epi_10_CYSTM1 cell cluster and Tregs, which might help to establish an immunosuppressive TIME and sustain heightened stemness in tumor cells.

## Discussion

CC is often regarded as less aggressive compared to ovarian cancer and endometrial cancer, primarily due to the effectiveness of HPV vaccines in prevention. When diagnosed at early stages, the 5-year survival rate for CC patients exceeds 90%, surpassing that of other gynecologic malignancies (*Gennigens et al., 2022*; *Schiffman et al., 2007*). However, if CC metastasizes to the intraperitoneal or retroperitoneal LNs, the survival rate will decrease to 30–60% (*Cohen et al., 2019*). Worse still, the 5-year survival rate for recurrent CC patients is less than 20% (*Li et al., 2016*). Specifically, when considering the ADC type alone, the prognosis is even worse. Unfortunately, our understanding for ADC is limited. Most previous studies have concentrated on SCC type of CC, and obtained heterogenetic information

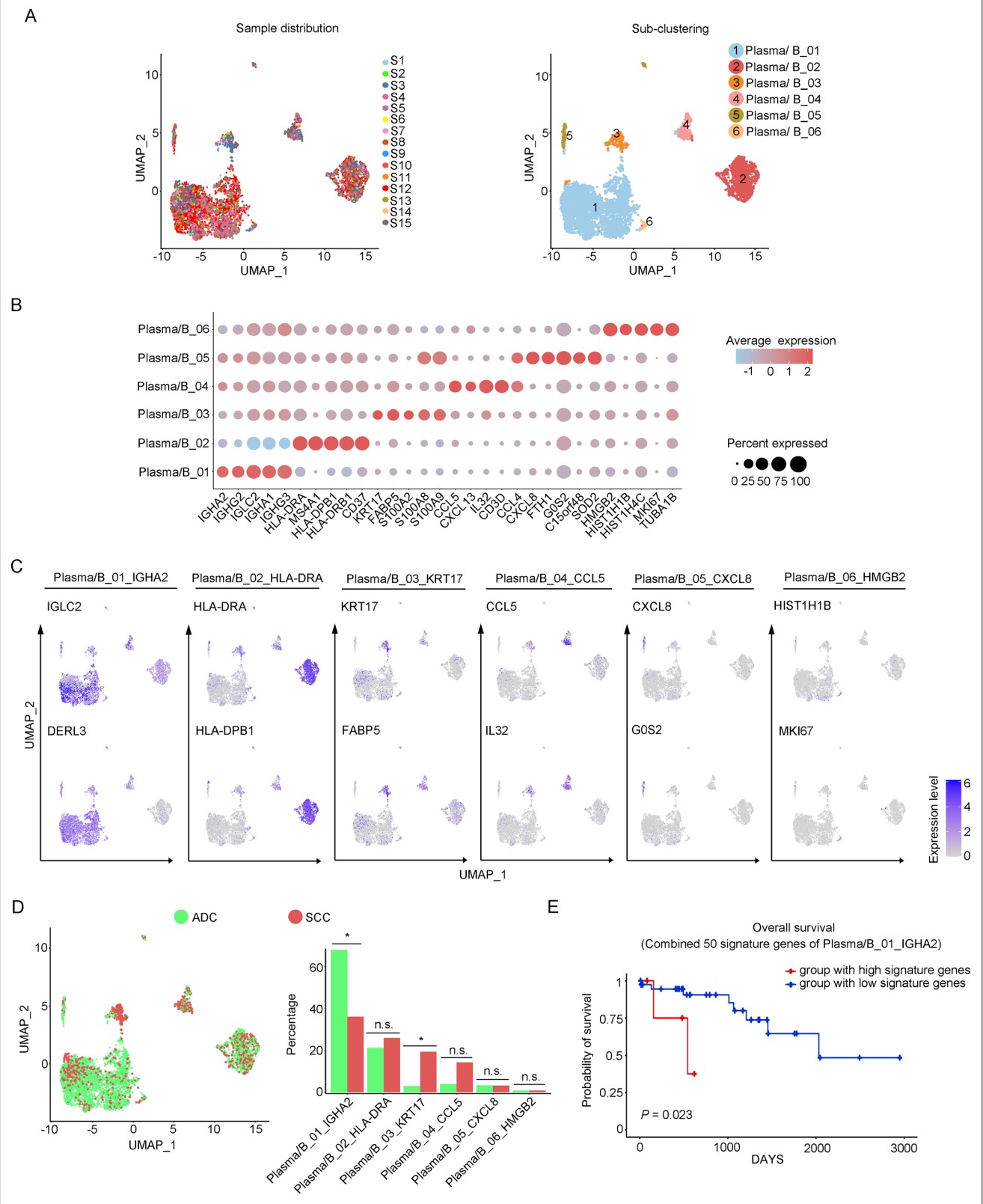

**Figure 6.** Phenotype diversity of plasma/B cells in adenocarcinoma (ADC). (**A**) Uniform Manifold Approximation and Projection (UMAP) plots showing the distribution of plasma/B cells in 15 samples (left) and the re-clustering into six clusters (right) according to ectopic expressions of genes. (**B**) Plot heatmap showing the annotation of each sub-cluster with top 5 differently expressed genes (DEGs). (**C**) UMAP plots demonstrating the most specific genes as the marker for each sub-cluster. (**D**) UMAP and histogram plots to compare the differences of distribution and proportion of each plasma/B

*Figure 6 continued on next page*

*Figure 6 continued*

cell sub-cluster between different histological types. Statistics were performed using R software with two-sided Wilcoxon test (p values for each group are listed below: Plasma/B_01_IGHA2: p=0.040; Plasma/B_02_HLA-DAR: p=0.177; Plasma/B_03_KRT17: p=0.040; Plasma/B_04_CCL5: p=0.388; Plasma/B_05_CXCL8: p=0.169; Plasma/B_06_HMGB2: p=0.211). Statistics are shown as *p<0.05; **p<0.01; n.s., not significant. (**E**) Kaplan–Meier curve showing the overall survival rate of CC patients stratified by the top 50 genes-scaled signature of Plasma/B_01_IGHA2.

The online version of this article includes the following source data and figure supplement(s) for figure 6:

**Source data 1.** Annotated code for data used in *Figure 6*.

**Figure supplement 1.** Phenotype diversity of plasma/B cells in adenocarcinoma (ADC).

derived from bulk RNA-seq method. Therefore, it is worthwhile to shift our attention toward an in-depth investigation on the disparities between ADC and SCC at the single-cell resolution.

More recently, researchers have started to utilize scRNA-seq approach to investigate the tumor microenvironment of CC. For instance, *Cao et al., 2023* compared five cases of SCC tumor tissues with paired adjacent normal tissues and described the immune landscape in which specific clusters of T and B cells exhibited immune-exhausting or activating processes. Another study conducted by *Qiu et al., 2023* aimed to elucidate the distinct molecular patterns of immune reactions between ADC and SCC in the context of different HPV infection status at the single-cell level. However, due to limited samples tested in these studies, our knowledge on the TIME in CC is yet inadequate. In the current study, we have generated a genomic atlas of ADC, which may depict the landscape of TIME built by both ADC tumor cells and neighboring cells. Apart from the predominant presence of epithelial cells, the cell clusters in ADC are majorly composed of T cells, tumor-associated neutrophils (TANs), and plasma/B cells. The gene subsets identified within ADC-abundant cell sub-clusters, including Epi_10_CYSTM1, Tregs, pro-tumor TANs, and P/B_01_IGHA2, exert potential roles in promoting both the carcinogenesis and immunosuppression of ADC. In the TIME of ADC, the unique module of cellular interaction has preliminarily indicated that the immunosuppressive Tregs are recruited to tumor cells via chemo-attraction. Finally, our hypothesis from scRNA-seq data is further confirmed using clinical samples, providing evidence that SLC26A3 may be a potential biomarker for ADC (*Figure 7—figure supplement 2*).

It has been reported that immunotherapies are less effective in ADC patients, especially those who are not infected with HPV (*Attademo et al., 2020*; *Ferrall et al., 2021*). As the two major components of ADC cell types in our study, T cells and epithelial cells have been implied to be the dominant regulators in the TIME due to their reciprocal interaction dynamics. We identified several ADC-specific epithelial cell clusters that may exhibit stronger stemness and aggressiveness than other clusters. Among these clusters, Epi_10_CYSTM1 seemed to have the highest level of malignancy, but the underlying mechanism behind this behavior remains unknown. The CellChat analysis revealed that Tregs might quite actively crosstalk to Epi_10_CYSTM1. To confirm this, immunostaining for co-localization demonstrated an increased aggregation of Tregs around tumor cells exhibiting overexpression of SLC26A3. It has also been widely reported that tumor cells acquire higher stemness present more aggressive features of resistance to immunotherapy (*Dianat-Moghadam et al., 2022*). This study indicated that TGFβ produced by Tregs might induce the stemness of Epi_10_CYSTM1 cells by activating downstream oncogenic pathways, inducing EMT on tumor epithelial cells. Because TGFβ is a soluble protein and is hard to detect on solid tissue samples, here we only validated markers of EMT and stemness to indirectly indicate the functional existence of TGFβ, which is predicted from informatic hypothesis (*Nixon et al., 2023*). Interestingly, Epi_10_CYSTM1 cells express ALCAM, which is a membranal ligand, and bind to its receptor CD6 on Tregs, activating chemotaxis and facilitating Treg recruitment (*Figure 7I*). The infiltration of a large number of Tregs in ADC may significantly accelerate immune evasion (*Figure 7—figure supplement 2*). Therefore, the presence of a higher number of Tregs infiltrating the tumor area of ADC may partially explain the increased resistance to immunotherapy.

Another interesting phenomenon observed in our sing-cell sequencing data is the involvement of TANs in the regulatory interactions within the TIME of ADC. It has been clearly investigated that pro-tumor TANs, characterized by CD66, CD11, CD170, and PD-L1, exert the oncogenic roles by promoting tumor cell proliferation, angiogenesis, inducing genetic instability, and most importantly causing immunosuppression (*Jaillon et al., 2020*). In this study, pro-tumor TANs are relatively enriched

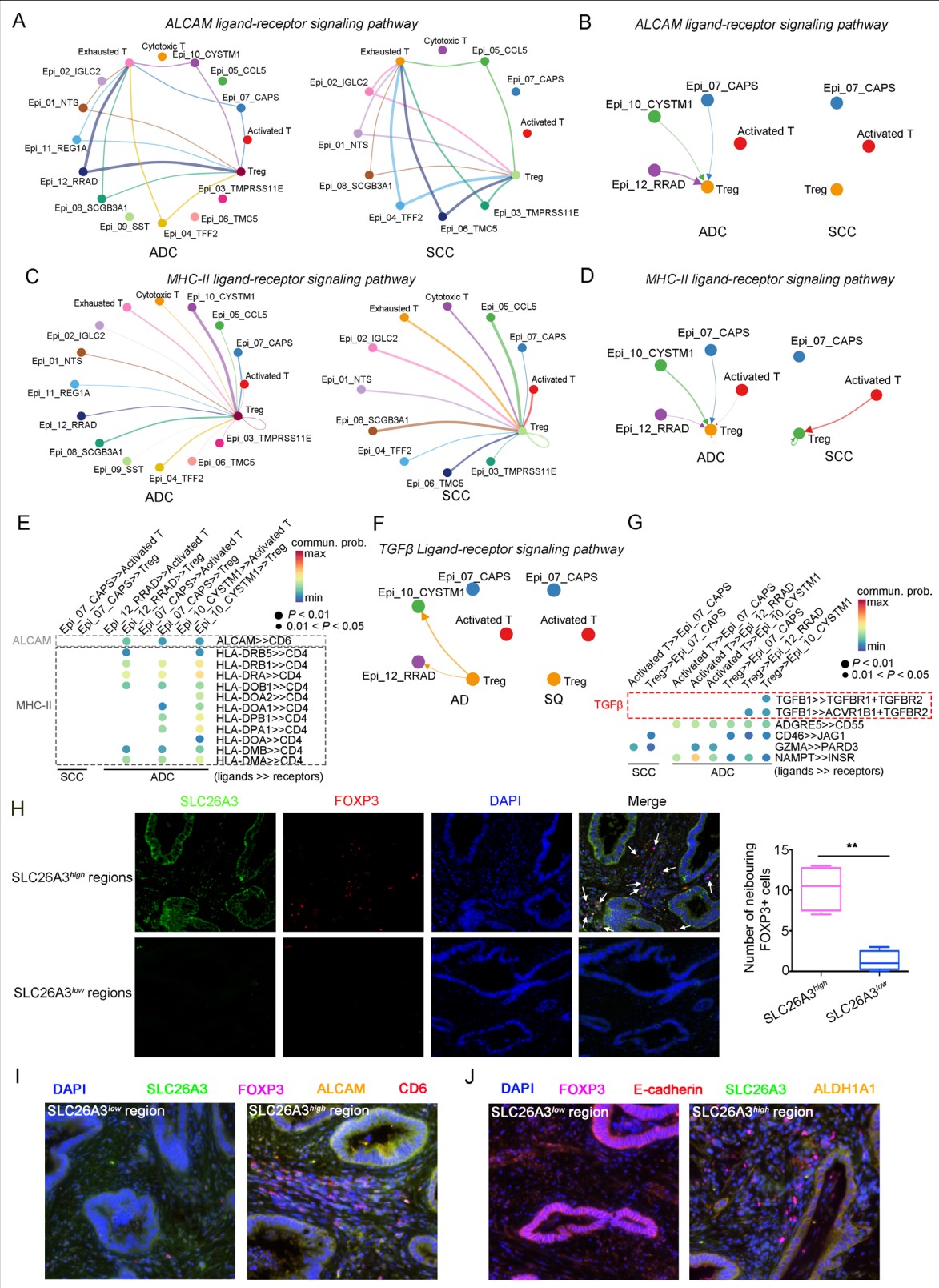

**Figure 7.** The cellular interaction modules of sub-clusters of T cells, neutrophils, and tumor epithelial cells. (**A**) Circle plots showing the interacting networks between epithelial cell sub-clusters and T cell sub-clusters via the pathway of ALCAM by comparing adenocarcinoma (ADC) and squamous cell carcinoma (SCC). (**B**) Circle plots simplified from (**A**) to show the interactions among target cell clusters via ALCAM pathway. (**C**) Circle plots showing the interacting networks between epithelial cell sub-clusters and T cell sub-clusters via the pathway of MHC-II by comparing ADC and SCC. (**D**) Circle

*Figure 7 continued on next page*

*Figure 7 continued*

plots simplified from (**C**) to show the interactions among target cell clusters via MHC-II pathway. From (**A**) to (**D**), the direction of each arrow shows the regulation from outputting cells to incoming cells. The width of the each line shows the predicted weight and strength of regulation. (**E**) Bubble plot showing the probability of ligand-to-receptor combination of each pathway between two different target sub-clusters of cells by comparing ADC with SCC. (**F**) Circle plots showing Tregs regulate epithelial cells via the TGF-β pathway, which is solely activated in ADC. (**G**) Bubble plot showing the probability of ligand-to-receptor combination of TGF-β pathway between Tregs and epithelial cells by comparing ADC with SCC. The pathways of ADGRE5, CD46, GZMA, and NAMPT are used as negative controls. (**H**) Dual immunofluorescence (IF) staining confirming that in SLC26A3$^{high}$ regions of CC tissues, more FOXP3$^+$ cells are recruited than in SLC26A3$^{low}$ regions (left). The numbers of recruited FOXP3$^+$ cell are quantified using histogram plot (right). Three individual samples with ROI were calculated and p<0.01 were marked with **, showing significant difference. (**I**) Multiplexed IF staining confirming the interaction between CD6 (on FOXP3$^+$ cells) and ALCAM (on SLC26A3$^{high}$ epithelial cells) in the ALCAM pathway. (**J**) Multiplexed IF staining showing that the recruitment of FOXP3 + cells toward SLC26A3$^{high}$ cells might induce EMT (marked with E-cadherin) and increase the stemness (marked with ALDH1A1) of tumor cells, via TGF-β pathway.

The online version of this article includes the following figure supplement(s) for figure 7:

**Figure supplement 1.** The cellular interaction modules of sub-clusters from T cells, neutrophils, and tumor epithelial cells.

**Figure supplement 2.** Graphical presentation of the tumor immune microenvironment (TIME) in adenocarcinoma (ADC), with cellular crosstalk among tumor cells, Tregs, neutrophils, and plasma cells.

in ADC tissues compared to SCC, especially in HPV-negative ADC cases. This suggests that ADC-specific epithelial cell sub-clusters may actively engage in the recruitment of pro-tumor TANs.

The anti-tumor function of B cells has been identified previously (***Bod et al., 2023***; ***Helmink et al., 2020***). However, our study has presented a specific cluster of plasma/B cells with tumor-promoting functions. The overexpression of gene signatures from Plasma/B_01_IGH2 cells is associated with poor regression in CC patients (***Figure 6F***). Interestingly, Plasma/B_01-IGHA2 was found to be closely related to the Plasma cell Cluster (PC Cluster) in Cao's study due to the shared gene set (***Cao et al., 2023***), such as IGHA1, IGHA2, IGHG2, and IGHG4 However, their roles were contradictory. According to that study, PC cluster was derived from the local expansion of TIME and showed potential effectiveness in anti-tumor immunity. In our case, Plasma/B_01-IGHA2 was indicated to have a reverse effect and played a tumor-promoting role in ADC. This difference in function is likely attributed to the distinct cell types and their interactions in ADC, instead of SCC. However, the underneath dynamics of B cells in ADC still needs further investigation.

Owing to post-surgical upstaging, some CC patients with stage IIIC are misdiagnosed as early-stage and have undergone radical surgeries. For these patients, the most appropriate strategy of treatment should be radiotherapy alone. However, it is challenging to avoid this issue because it is difficult to identify metastatic LN only through CT or MRI. To solve this problem, a reliable biomarker is required to diagnose LN metastasis in conjunction with radiographic tool. In this study, SLC26A3 has shown its potential as a diagnostic biomarker to predict LN metastasis in ADC patients. Combining two large cohorts of clinical data, we found that CC patients with ADC types were more vulnerable to be misdiagnosed for clinical staging. Derived from scRNA-seq data, SLC26A3 could be employed as a representative gene for the Epi_10_CYSTM1 cluster, which was found to be enriched in stage IIIC cases of ADC. The IHC staining further confirmed that ADC patients at stage IIIC exhibited a higher expression of SLC26A3 compared to early stages (FIGO stages I–II). Furthermore, we tested on biopsy samples and it implied that SLC26A3 might serve as a potential predictor of LN metastasis. Our study for the first time provides evidence that ADC patients are at a higher risk of developing LN metastasis, which can be challenging to detect solely via radiological imaging techniques. However, by incorporating LN imaging with biomarkers such as SLC26A3, it may probably reduce the misdiagnosis rate among FIGO IIIC patients.

In this study, our objective is to unravel the genomic characteristics of ADC in the hope of identifying gene signatures that can elucidate its aggressive nature. SLC26A3 is selected from the DEGs enriched in cluster Epi_10_CYSTM1, and is identified to specifically represent this ADC-specific cluster. However, based on our data, we have inevitably encountered two questions: (1) the existence of cluster Epi_10_CYSTM1 and (2) the specificity of SLC26A3. As for the first question, Epi_10_CYSTM1 is distinctive from the other 11 epithelial cell clusters, with specific DEGs (***Figure 2A–C***). Although Epi_10_CYSTM1 is dominantly derived from sample S7, however, its proportion is relatively high when compared with other clusters such as Epi_08_SCGB3A1, Epi_09_SST, and Epi_11_REG1A (***Figure 2D***, ***Figure 2—figure supplement 1A and B***). Moreover, we have detected a positive expression of this

cluster on tissue samples, whether by using ORM1/2 or SLC26A3 antibodies (two candidate markers of cluster Epi_10_CYSTM1) via IHC. As a result, we have identified the existence of this cell cluster. As for the second question, SLC26A3 has demonstrated a higher specificity compared to other candidates, such as CYSTM1, CA9, ORM1, and ORM2, following the process of identifying markers (*Figure 3A and B*, *Figure 3—figure supplement 1E*). Although the IHC results indicate that SLC26A3 is positively expressed in stage IIIC cases, however, the expression is confined to specific regions (*Figure 3—figure supplement 2*), which aligns with the scRNA-seq findings for SLC26A3. Notably, according to *Tables 3 and 4*, most of the clinical cases have a lower expression of SLC26A3. Taken together, SLC26A3 is a specific representative of cluster Epi_10_CYSTM1.

In summary, we characterized the genomic landscape of the TIME in ADC at a single-cell resolution. Our study revealed the presence of specific epithelial cell clusters in ADC that exhibited more aggressive features. The enriched gene signatures from these clusters may serve as potential therapeutic targets to improve the efficacy of immunotherapy. The interactions between tumor epithelial cells and other types of cells, such as T cells and neutrophils, appropriately depict a highly immunosuppressive microenvironment of ADC. Moreover, our investigation extended to the issue of post-surgical upstaging at the single-cell level, with the identification of biomarkers from specific tumor cell clusters showing potential as diagnostic indicators for the precise diagnosis of CC patients.

# Materials and methods

## Clinical sample collection

The 15 cases of fresh CC tissues were collected from Hunan Cancer Hospital, with informed consent obtained from 15 independent patients. The inclusion criteria are as follows: (1) the diagnosis of CC, with pathological type of ADC or SCC, should be confirmed by pathologists; (2) pre-surgical FIGO stages should be within the range from IB (which means the tumor mass should be detectable via CT or MRI) to IIA (which means no parametrial involvement, nor distant metastasis), in which patients have explicit indication for radical surgery following NCCN guidelines; (3) no neo-adjuvant treatments should be given before the surgery, including chemotherapy and radiotherapy; (4) the surgery options should be type C radical hysterectomy and include pelvic lymphadenectomy with/without para-aortic lymphadenectomy. Detailed information of the 15 patients is listed in *Supplementary file 1*. This study was supervised and approved by the Ethics Committee Board of Hunan Cancer Hospital.

## Sample preparation for single-cell isolation

Immediately after surgical removal, the fresh tumor tissues were washed by saline for three times and stored in cold storage solution (MACS Tissue Storage Solution) for transportation. To prepare high-quality samples, the tissues were cut into pieces to 1 mm$^3$ and were incubated at 37°C for 30 min. 0.25% trypsin solution was used to digest the tissue pieces at 37°C for 10 min. Then single-cell samples were filtered through a 70 μm meshed filter and suspended in a red blood cell lysis buffer (MACS Red Blood Cell Lysis Solution) for dissolution. Before instrumental sequencing, suspension of single cells were visualizing by TC20 Automated Cell Counter to assess the cell viability. Each sample was confirmed to have a cell viability rate of over 80%.

## Single-cell RNA sequencing

The suspension of single cells was transferred onto the 10X Chromium Single-Cell instrument to generate single-cell beads in the emulsion (GEMs). Then the scRNA-seq libraries were constructed by using the Chromium Controller and Chromium Single Cell 3′Reagent Kits (v3 chemistry CG000183) and sequenced by using the sequencer Novaseq6000 (Illumina, USA). All procedures followed the manufacturer's protocol.

## Data processing, dimension reduction, and cell clustering

The raw scRNA-seq reads were processed for barcode processing, genome mapping, and the gene expression matrixes were generated by using the Cell Ranger toolkit (version 5.0.0) of 10X Genomics platform. The GRCh38 human reference genome was utilized in the read alignment process. The unique molecular identifiers (UMIs) were counted in each cell. As for data processing, cells with low qualities, which showed <200 expressed genes or >25% mitochondrial UMIs, were excluded. The

package of Seurat R (version 4.0.5R) was applied for quality control. The Scrublet software (version 0.2.2) was utilized to identify and remove potential doublets. As for data normalization and conversion, the LogNormalize method was implemented in the NormalizeData function, and then the normalized counts were log-transformed. With the function of RunPCA, the principal component analysis (PCA) was used for data dimension reduction. The functions of FindNeighbors and FindClusters were used for cell clustering. For visualization, the RunUMAP function was performed. In each unsupervised cell cluster, the gene markers were identified by the function of FindAllMarkers with comparison to other clusters.

## Cell type annotation

The cell typing was conducted and annotated according to the selected gene markers via CellMarker database and publications (*Li et al., 2022*; *Qiu et al., 2023*; *Xue et al., 2022*). The expression of DEGs was used to determine each cell type following the rules: $Log_2Foldchange$ should be >0.2 and adjusted p-values should be <0.05.

## Pseudotime trajectory analysis

To identify the developmental changes of epithelial cells, the pseudotime trajectory analyses were performed by using Monocle2 R package (version 2.22.0), in which Seurat was used as input. Calculated using the Monocle algorithm, the top 50–1000 genes were conducted for differential GeneTest. Then, the DDR-Tree and default parameters were generated to visualize the translational relationship and developmental orders among different epithelial sub-clusters.

## Cell malignancy scoring

Among these epithelial cells, the malignancy score of each single cell was predicted by using scCancer R package (*Guo et al., 2021*). The malignant cells were distinguished from non-malignant cells by inferring large-scale copy number variations, which was calculated by the interCNV R package as described (*Guo et al., 2021*; *Xue et al., 2022*).

## Signaling pathway enrichment analysis

The biological signaling pathway of each cell cluster was identified by performing Gene Ontology (GO) analyses and Hallmark Pathway enrichment analyses, according to the Molecular Signature Database (MSigDB version 7.5.1). Then DEGs of each cell cluster were converted to ClusterProfiler (version 4.2.2) package for the analyses of predicted functions.

## CytoTRACE

In order to predict the differential state of cells, scRNA-seq-derived data were passed to the CytoTRACE software (version 0.3.3) with R package for further analyses.

## Survival analysis

The CESC dataset from the TCGA database was downloaded from Xena, with clinical characteristics, survival endpoints, and treatments, was included. The Kaplan–Meier survival curves were generated using R software.

## Cellular communication analysis

The possible interactions, with ligand-to-receptor communicating signaling pathway, between two types of cells were analyzed using the R packages of CellChat (version 1.5.0). Seurat was used for normalization, and the rank with significance was calculated. The probability of ligand-to-receptor interaction in each signaling pathway was plotted by p-value and intensity.

## Immunohistochemistry

The paraffin-embedded ADC samples (whether post-surgical or biopsy samples) were sectioned into 4 μm slides. The protocol of IHC strictly followed our previous published works (*Peng et al., 2019*). The primary antibody of SLC26A3 was purchased from Santa Cruz Biotechnology (Cat# sc-376187) and diluted at the ratio of 1:20. The protein expression of SLC26A3 was determined using the method of *H*-scoring: *H*-score = intensity score × percentage of positivity. The intensity was scored as negative

= 0, weak = 1, moderate = 2, strong = 3. Samples were grouped as the higher expression group (*H-score* ≥150) and the lower expression (*H-score* <150) group of SLC26A3.

## Multiplexed immunofluorescence (multi-IF)

The multi-IF was performed using the Opal 6-Plex Manual Detection Kit (Akoya Biosciences, NEL811001KT), following the manufacturer's instructions. Generally, the slides were stained with each primary antibody sequentially following the staining steps in each cycle as (1) antigen retrieval by citrate (pH = 6) at water bath for 30 minutes; (2) independent primary antibody incubation at 4°C atmosphere overnight or 37°C atmosphere for 1 hour; (3) common HRP-crosslinked secondary antibody incubation at 37°C atmosphere for 30 minutes; and (4) independent opal reactive fluorophore solution at intended wavelength (opal 570 was excluded in order to save for FOXP3 signal). Each independent primary antibody started a new cycle. After all cycles finished, the slides were prepared for incubation of FOXP3 monoclonal antibody with eFluor 570 (Thermo Fisher, Cat#41-4777-82) conjugated, as well as DAPI (Sigma-Aldrich, Cat#28718-90-3) for nucleus staining.

## Statistics

Specific analyses of scRNA-seq data were decoded and visualized using the R software. Statistics were processed using IBM-SPSS Statistics 20.0 software and R software as well, including Student's *t*-test, two-sided Wilcoxon test, etc. Correlations analysis was performed using the $\chi^2$ test. Survival analysis was performed using the Kaplan–Meier method. p values <0.05 were considered statistically significant.

## Additional information

### Funding

| Funder | Grant reference number | Author |
|---|---|---|
| National Natural Science Foundation of China | 82002753 | Yang Peng |
| Natural Science Foundation of Hunan Province | 2021JJ40324 | Yang Peng |
| National Natural Science Foundation of China | 82072882 | Congrong Liu |
| Natural Science Foundation of Hunan Province | 2022JJ70103 | Chaonan Chu |
| National Natural Science Foundation of China | 81500475 | Jing Yang |

The funders had no role in study design, data collection and interpretation, or the decision to submit the work for publication.

### Author contributions

Yang Peng, Conceptualization, Data curation, Formal analysis, Funding acquisition, Investigation, Visualization, Methodology, Writing – original draft, Project administration; Jing Yang, Data curation, Formal analysis, Investigation, Methodology, Writing – review and editing; Jixing Ao, Resources, Data curation, Software, Validation, Investigation, Methodology; Yilin Li, Resources, Data curation, Validation, Investigation, Visualization, Methodology; Jia Shen, Data curation, Software, Validation, Investigation, Visualization; Xiang He, Data curation, Software, Investigation, Visualization; Dihong Tang, Formal analysis, Methodology, Writing – review and editing; Chaonan Chu, Formal analysis, Funding acquisition, Writing – review and editing; Congrong Liu, Conceptualization, Formal analysis, Supervision, Funding acquisition, Methodology, Project administration, Writing – review and editing; Liang Weng, Conceptualization, Software, Formal analysis, Supervision, Validation, Investigation, Methodology, Writing – original draft, Project administration, Writing – review and editing

## Author ORCIDs
Congrong Liu ![ORCID] http://orcid.org/0000-0002-9876-1603
Liang Weng ![ORCID] https://orcid.org/0000-0001-7058-8274

## Ethics
Specimen collection and pathological data used in this study were approved by the Ethics Committee of Hunan Cancer Hospital (No. KYJJ-2020-036). Written informed consent was obtained from patients for use of samples. This study was also in accordance with the Declaration of Helsinki and none of any procedures conducted interfered with the treatment plan of patients.

Reviewer #1 (Public review): https://doi.org/10.7554/eLife.97335.3.sa1
Reviewer #2 (Public review): https://doi.org/10.7554/eLife.97335.3.sa2
Author response https://doi.org/10.7554/eLife.97335.3.sa3

---

# Additional files

## Supplementary files
Supplementary file 1. Table of sample information from the 15 individual patients.

Supplementary file 2. Table of composition of cell types in the scRNA-seq data.

Supplementary file 3. Table of gene markers for cell type identification.

MDAR checklist

## Data availability
In this study, the single-cell RNA sequencing data generated is available in the NCBI Sequence Read Archive with accession number: PRJNA1231266.

The following dataset was generated:

| Author(s) | Year | Dataset title | Dataset URL | Database and Identifier |
|---|---|---|---|---|
| Peng Y, Yang J, Ao J, Li Y, Shen J, He X, Tang D, Chu C, Liu C, Weng L | 2024 | Single-cell profiling reveals the intratumor heterogeneity and immunosuppressive microenvironment in cervical adenocarcinoma | https://www.ncbi.nlm.nih.gov/bioproject/PRJNA1231266/ | NCBI BioProject, PRJNA1231266 |

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

# Appendix 1

## Appendix 1—key resources table

| Reagent type (species) or resource | Designation | Source or reference | Identifiers | Additional information |
|---|---|---|---|---|
| Antibody | SLC26A3 Antibody (H-8) | Santa Cruz Biotechnology | Cat# sc-376187 | |
| Antibody | FOXP3 Monoclonal Antibody (236A/E7), eFluor 570, eBioscience | Thermo Fisher | Cat# 41-4777-82 | |
| Antibody | ALCAM polyclonal antibody | Proteintech | Cat# 21972-1-AP | |
| Antibody | CD6 recombinant antibody | Proteintech | Cat# 84508-4-RR | |
| Antibody | ALDH1A1 polyclonal antibody | Proteintech | Cat# 15910-1-AP | |
| Antibody | Anti-IGF2 antibody | Abcam | Cat# ab9574 | |
| Antibody | ADH1C rabbit pAb | ABclonal | Cat# A8081 | |
| Antibody | ORM1/2 rabbit mAb | ABclonal | Cat# A19736 | |
| Antibody | E-Cadherin (24E10) rabbit mAb | Cell Signaling Technology | Cat# 3195 | |
| Peptide, recombinant protein | Opal 6-Plex Manual Detection Kit | Akoya Biosciences | Cat# NEL811001KT | |

