## [Editor Report · eLife Assessment]

In this **useful** article, the authors performed scRNA-seq on a diverse cohort of 15 early-stage cervical cancer patients. Correlative data is provided to support the possible establishment of an immunosuppressive microenvironment near SCL26A3+ cells, and an association of these cells with upstaging at time of surgery. However without more extensive validation, the evidence supporting the conclusions remains **incomplete**. Overall, this article will provide a potentially helpful dataset for researchers studying cervical cancer.

---

## [Referee Report · Reviewer #1 (Public review)]

Summary:

The authors in this manuscript performed scRNA-seq on a cohort of 15 early-stage cervical cancer patients with a mixture of adeno- and squamous cell carcinoma, HPV status, and several samples that were upstaged at the time of surgery. From their analyses they identified differential cell populations in both immune and tumour subsets related to stage, HPV status, and whether a sample was adenocarcinoma or squamous cell. Putative microenvironmental signaling was explored as a potential explanation for their differential cell populations. Through these analyses the authors also identified SLC26A3 as a potential biomarker for later stage/lymph node metastasis which was verified by IHC and IF. The dataset is likely useful for the community. The accuracy and clarity have been improved from the previous version, and additional immunofluorescence supporting the existence of their proposed cluster is now present. That said, there remain some issues with the strength of some claims (particularly in the abstract and results sections) and some of the cell type definitions.

Strengths

The dataset could be useful for the community

SLC26A3 could potentially be a useful marker to predict lymph node metastasis with further study

Weaknesses

Casual language is used in the abstract around immunosuppressive microenvironment and signal cross-talk between Epi_10_CYSTM1 cluster and Tregs. The data show localization that supports a possible interaction and probable cytokines, but functional experiments would be needed to establish causality.

In the description of the single cell data processing there is no mention of batch effect correction. Given that many patients were analyzed, and no mention was made of pooling or deconvolution, it must be assumed these were run separately which invariably leads to batch effects. Given the good overlays across patients some batch correction must have been performed. How was batch effect correction performed?

While statistics were added to the clinical correlates, it would appear that single variables are being assessed one at a time by chi-squared analysis. This ignores the higher order structure of the data and the correlations between some variables resulting in potentially spurious findings. This is compounded as some categories had below 5 observations violating the assumptions of a chi-squared test.

The description of all analytical steps remains quite truncated. While the inclusion of annotated code is useful, a full description of which tools were used, with which settings, and why each were chosen, is a minimum needed to properly interpret the results. This is as important in a mainly analytical paper as the experimental parameters.

Validation of the clustering results remains a problem. The only details provided are that FindClusters was used. This depends on a manual choice of multiple parameters including the k-nearest neighbours included, whether Louvain or Leiden clustering is used, the resolution parameter, and others (how many variable genes/PCs etc...). Why were these parameters selected, how do you know that you're not over or under-clustering.

The cluster Epi_10_CYSTM1 remains somewhat problematic. None of the additional data supports its existence outside of the single patient who has cells from that population. Additionally, it falls well outside of any of the other Epithelial cells to the point that drawing it as part of a differentiation order doesn't even make sense. Indeed, most of the upregulated pathways in this cluster appear to be related to class II antigen presentation which would fit better with a dendritic cell/macrophage than an epithelial cell. While the IF at the end does support the existence of the cluster, numbers are still very limited, and this doesn't have data on the antigen presenting function. At the least a strong disclaimer should be included in the text that this population is essentially exclusive to one sample in the scRNA data.

The linkage between the cluster types and IHC for prediction of lymph node metastasis is tenuous. Most of the strongly cluster associated markers were not predictive despite their clusters being theoretically enriched. This inability to recognize the clusters in additional samples using alternative methods does not give confidence that these clusters are robust. SLC26A3 being associated with upstaging may very well be a useful marker, however, given the lack of association of the other markers, it may be premature to say this is due to the same Epi_10_CYSTM1 cluster.

There are multiple issues in the classification of T cells and neutrophils. In the analysis of T cell subset, all CD4+ T cells are currently scored as Tregs, what happened to the T-helper cells? Additionally, Activated T and Cytotoxic T both seem to contain CD8+ cells, but all their populations have equivalent expression of the activation marker CD69. Moreover, the "Cytotoxic" ones also express TIGIT, HAVCR2 and LAG3 which are generally exhaustion markers. For neutrophils, several obviously different clusters have been grouped together (Neu_1 containing two diametrically opposite cell clouds being an obvious example).

Again in the CellChat section of the results causal language is being repeatedly used. These are just possible interactions, not validated ones. While the co-localization in the provided IF images certainly supports the co-localization, this still is only correlative and doesn't prove causality.

Minor Issues

The sentence "However, due to the low morbidity of ADC, in-depth investigations are insufficient" could be misinterpreted. Morbidity generally refers to the severity or health burden rather than the frequency of cases, though it's true in some studies prevalence is used for the overall impact of the disease on a population and referred to as morbidity. In this instance though, "incidence" or "prevalence" would be clearer word choices.

The previous rebuttal states that clusters/cell type calls were refined to eliminate issues such as epithelial cells creeping into the T cell cluster, however, the cell %s have not been altered according to the change tracking. Shouldn't all the %s have been altered even if only slightly?

---

## [Referee Report · Reviewer #2 (Public review)]

Summary:

Peng et al. present a study using scRNA-seq to examine phenotypic properties of cervical cancer, contrasting features of both adenocarcinomas (ADC) and squamous cell carcinoma (SCC), and HPV-positive and negative tumours. They propose several key findings: unique malignant phenotypes in ADC with elevated stemness and aggressive features, interactions of these populations with immune cells to promote an immunosuppressive TME, and SLC26A3 as a biomarker for metastatic (>=Stage III) tumours.

Strengths:

This study provides a valuable resource of scRNA-seq data from a well-curated collection of patient samples. The analysis provides a high-level view of the cellular composition of cervical cancers. The authors introduce some mechanistic explanations of immunosuppression and the involvement of regulatory T cells that is intriguing.

Weaknesses:

I believe many of the proposed conclusions are over-interpretations or unwarranted generalizations of the single-cell analysis. I believe there may also be some artifacts in the data that may not reflect true biology--eg. The presentation of KRT+ neutrophils, which may reflect doublets with cancer cells. In some cases there is mention of quality control steps to remove contaminant cell clusters, but there is no method or supplemental figure to describe and/or justify these steps.

The key limitation is related to the "ADC-specific" Epi_10_CYSTM1 cluster, which is a central focus of the paper. This population only contains cells from one of the 11 ADC samples and represents only a small fraction of the malignant cells from that sample. Yet, this population is used to derive SLC26A3 as a potential biomarker. SLC26A3 transcripts are only detected in this small population of cells (none of the other ADC samples), which makes me question the specificity of the IHC staining on the validation cohort. The manuscript does not address why this marker is so rare in the scRNA-seq data, but abundant in the IHC.

While I understand it may be out of the scope of this individual study, many of the conclusions are inferred from the data analysis with little follow-up in experimental models or orthogonal assays.

---

## [Author Response]

The following is the authors’ response to the original reviews.

**Public reviews:**

**Reviewer #1 (public review):**
(1) The link between the background in the introduction and the actual study and findings is often tenuous or not clearly explained. A re-working of the intro to better set up and link to the study questions would be beneficial.

We have rewritten the introduction of the manuscript and clearly stated the study questions we were aiming for:

In paragraph 1-we have stated clearly that we need to study why ADC type of cervical cancer is more aggressive. (Line 58 - 77)

In paragraph 2- we have stated clearly that we need to find valuable biomarkers to help diagnose lymph node metastasis, which may compensate the shortage of radiological imaging tools and reduce the rate of misdiagnosis. (Line 78 - 100)

In paragraph 3- we have stated clearly that HPV negative cases is a special group of cervical cancer and we aim to study its cellular features. (Line 101 - 108)

In paragraph 4- we have stated clearly that we need to decode cell-to-cell interaction mode in the tumor immune microenvironment of ADC using scRNA-seq. (Line 109 - 123)

(2) For the sequencing, which kit was used on the Novaseq6000?

For sequencing, we used the Chromium Controller and Chromium Single Cell 3’Reagent Kits (v3 chemistry CG000183) on the Novaseq6000. We feel sorry for lacking this quite important part and have already add the information in Methods section. (Line 196- 197)

(3) Additional details are needed for the analysis pipeline. How were batch effects identified/dealt with, what were the precise functions and settings for each step of the analysis, how was clustering performed and how were clusters validated etc. Currently, all that is given is software and sometimes function names which are entirely inadequate to be able to assess the validity of the analysis pipeline. This could alternatively be answered by providing annotated copies of the scripts used for analysis as a supplement.

We apologize for the inadequacy of descriptions of data analysis process. We have already provided a new part of “data processing” with more details in the Methods section (Line 202 - 221). In addition, we have also provided annotated copies of scripts in the supplementary data as Supplementary Data 1.

(4) For Cell type annotation, please provide the complete list of "selected gene markers" that were used for annotation.

We have already added the list of marker genes for cell type annotation in the revised manuscript as Supplementary Table 3.

(5) No statistics are given for the claims on cell proportion differences throughout the paper (for cell types early, epithelial sub-clusters later, and immune cell subsets further on). This should be a multivariate analysis to account for ADC/SCC, HPV+/- and Early/Late stage.

We feel sorry for lacking statistics when performing analyses of comparisons. In the revision, we have already used statistic approaches to analyze the differences between each set of group comparison. As a result, the corresponding figures have been revised, accordingly.

For examle, Fig. 1F, Fig. 2D, Fig. 4E, Fig. 5D, Fig. 6D had been re-analyzed to compare ADC/SCC；Supplementary Fig. 1A, Supplementary Fig. 2A, Supplementary Fig. 4A, Supplementary Fig. 5A, Supplementary Fig. 6A had been re-analyzed to compare HPV+/HPV-; Supplementary Fig. 1B, Supplementary Fig. 2B, Supplementary Fig. 4B, Supplementary Fig. 5B, Supplementary Fig. 6B had been re-analyzed to compare Early/Late stage. All P values have been listed in the figure legends.

(6) The Y-axis label is missing from the proportion histograms in Figure 2D. In these same panels, the bars change widths on the right side. If these are exclusively in ADC, show it with a 0 bar for SCC, not doubling the width which visually makes them appear more important by taking up more area on the plot.

We feel sorry for impreciseness when presenting histograms of Fig. 2D and we have also revised other figures with similar mistakes, such as Fig. 1F, Fig. 5D. As for the width of bars, which is due to output style of data processing, we have already corrected all similar mistakes alongside the whole manuscript, for example, Fig. 2D and Supplementary Fig. 2A-B.

(7) Throughout the manuscript, informatic predictions (differentiation potential, malignancy score, stemness, and trajectory) are presented as though they're concrete facts rather than the predictions they are. Strong conclusions are drawn on the basis of these predictions which do not have adequate data to support. These conclusions which touch on essentially all of the major claims made in the manuscript would need functional data to validate, or the claims need to be very substantially softened as they lack concrete support. Indeed, the fact that most of the genes examined that were characteristic of a given cluster did not show the expected expression patterns in IHC highlights the fact that such predictions require validation to be able to draw proper inferences.

Thank you for your insightful comments. As you noted, several conclusions were initially based on bioinformatics predictions. Thus in the revised manuscript, we have rewritten all relevant descriptions in a more softened way, particularly in the paragraph of “epithelial cells” in Results section, as well as the conclusions derived from bioinformatics predictions in other paragraphs throughout the manuscript. We hope our revised descriptions will enhance the precision of our work.

For example, in paragraph “The sub-clusters of epithelial cells in ADC exhibit elevated stem-like features (from Line 353)”, many over-affirmative disriptions had been re-written in Line 353, 362, 371, 375, 379, 383, 390, 392. From Line 395 to 399, the conclusion had been revised as “The observation of cluster Epi_10_CYSTM1 and its possible specificity to ADC makes us question whether or not it may be related to the aggressiveness of ADC” compared to the previous “This observation may partially indicate that high stemness cluster Epi_10_CYSTM1 is essential for ADC to present more aggressive features”. From Line 400 to 408, conclusions from GO analyses had also been rewritten.

In paragraph “ADC-specific epithelial cluster-derived gene SLC26A3 is a potential prognostic marker for lymph node metastasis (from Line 422)”, many conclusions based on predictions had been revises, such as Line 424 - 428, Line 439 - 441, Line 451 - 453, Line 455 - 457, Line 458 - 459, Line 471 - 473, Line 478 - 481, Line 484 - 486, Line 489, etc.

In paragraph “Tumor associated neutrophils (TANs) surrounding ADC tumor area may contribute to the formation of a malignant microenvironment (from Line 536)”, we have changed the descriptions based on bio-infomative predictions, such as Line 560, Line 561, Line 565, Line 566, Line 572, Line 576 - 577, etc.

In paragraph “Crosstalk among tumor cells, Tregs and neutrophils establishes the immunosuppressive TIME in ADC (from Line 601)”, we have already corrected the all the affirmative descriptions, such as Line 604, Line 612, Line 614, Line 626, Line 628 - 629, Line 641, Line 654 – 655, etc.

All the changes have also been listed in Revision Notes in detail.

(8) The cluster Epi_10_CYSTM1 which is the basis for much of the paper is present in a single individual (with a single cell coming from another person), and heavily unconnected from the rest of the epithelial populations. If so much emphasis is placed on it, the existence of this cluster as a true subset of cells requires validation.

We appreciate this suggestion. We agree that the majority of Epi_10_CYSTM1 cells are derived from sample S7. The fact that we have detected this cluster in only one patient may be due to sampling differences and the inherent heterogeneity of tumor specimens. However, the relatively high number of cells in this cluster from one stage III patient suggests its presence in ADC patients and highlights its potential as a diagnostic marker for clinical staging. To further investigate whether this cluster is generally existing in ADC patients, we have identified and selected candidate genes, such as SLC26A3, ORM1, and ORM2, as representative markers of this cluster, which demonstrated high specificity (as shown in Fig. 3B). We then performed IHC staining on a total of 56 tissue samples, and the results showed positive expressions of these markers in the majority of stage IIIC tumor tissues, confirming the existence of this cell cluster (as shown in Supplementary Fig. 3E). In our revised manuscript, we have included an in-depth discussion of this issue in the seventh paragraph of the Discussion section (From Line 801).

(9) Claims based on survival analysis of TCGA for Epi_10_CYSTM1 are based on a non-significant p-value, though there is a slight trend in that direction.

Thank you for your insightful comment. From the data of TCGA survival analysis for Epi_10, we found a not-so-slight trend of difference between groups (with a small P value). As a result, we presented this data and hoped to add more strength to the clinical significance of this cluster. However, this indeed caused controversy because the P value is non-significant. As a result, we have already deleted this data in the revised manuscript.

(10) The claim "The identification of Epi_10_CYSTM1 as the only cell cluster found in patients with stage IIICp raises the possibility that this cluster may be a potential marker to diagnose patients with lymph node metastasis." This is incorrect according to the sample distributions which clearly show cells from the patient who has EPI_10_CYSTM1 in multiple other clusters. This is then used as justification for SLC26A3 which appears to be associated with associated with late stage, however, in the images SLC26A3 appears to be broadly expressed in later tumours rather than restricted to a minor subset as it should be if it were actually related to the EPI_10_CYSTM1 cluster.

We feel thankful for this question. The conclusion that “The identification of Epi_10_CYSTM1 as the only cell cluster found in patients with stage IIICp raises the possibility that this cluster may be a potential marker to diagnose patients with lymph node metastasis” has indeed been written too concrete according to the sample distribution. We feel sorry for this and have already corrected the description into “As one of stage IIIC-specific cell clusters, the cluster of Epi_10_CYSTM1, with its representative marker gene SLC26A3, presents potential diagnostic value to predict lymph node metastasis” from Line 478-481.

However, based on our results, we do think this cluster is a potential diagnostic marker and the hypothesis is right. As for SLC26A3, we have specifically added a new paragraph (from Line 801 - 822) in Discussion section to discuss the rationality and necessity of selecting this gene as our central focus, and the reasons why SLC26A3 should be the representative of cluster Epi_10_CYSTM1. As you noted, SLC26A3 appears to be broadly expressed in later tumors rather than restricted to a minor subset in the images. We apologize for any misunderstanding caused. When presenting the IHC data, we only showed the strongly positive areas of each slide to emphasize the differences. In our revision, we have included whole slide scanning images of the IHC samples, clearly showing that SLC26A3 is restricted to a part of the tumors (Supplementary Fig.9).

(11) The authors claim that cytotoxic T cells express KRT17, and KRT19. This likely represents a mis-clustering of epithelial cells.

We apologize for using data without noticing the contamination of T cells with few epithelial cells. We have re-performed quality control to exclude contamination and re-analyzed all data of T cells. In the reviesed manuscript, we have therefore updated completely new data for T cells in both Fig. 4 and Supplementary Fig. 4.

(12) Multiple claims are made for specific activities based on GO term biological process analysis which while not contradictory to the data, certainly are by no means the only explanation for it, nor directly supported.

Our initial purpose was to use GO analysis as supports for our conclusions. However, we know these are only claims but not evidence, which is also the problem of our writing techniques as in question (7). Therefore, in our revised manuscript, we have already deleted GO data and descriptions in the paragraphs of “T cell (Fig.4)”(from Line 495) and “B/plasma cell (Fig.6)” (from Line 579), because the predictions are quite irrelevant to our conclusions.

However, in the sections of “epithelial cell (Fig.2)” (from Line 352) and “neutrophils (Fig.5)” (from Line 536), we retained the GO data and rewrote the conclusions, because these analyses have provided us with valuable information regarding the role of specific cell clusters in ADC progression. Furthermore, our subsequent analyses, such as CellChat, have further validated the accuracy of the findings from the GO analysis. We do think this logically supports the whole storyline of the study.

**Reviewer #2 (public review):**
(1) I believe that many of the proposed conclusions are over-interpretations or unwarranted generalizations of the single-cell analysis. These conclusions are often based on populations in the scRNA-seq data that are described as enriched or specific to a given group of samples (eg. ADC). This conclusion is based on the percentage of cells in that population belonging to the given group; for example, a cluster of cells that dominantly come from ADC. The data includes multiple samples for each group, but statistical approaches are never used to demonstrate the reproducibility of these claims.

We feel sorry that many of the conclusions have been written in an over-affirmative way but lack profound supporting evidences. In our revision, we have already optimized the writing techniques and re-written all conclusions or descriptions related to only bio-informatic predictions. Moreover, we have performed statistical re-analyses on all data and rearranged the related figures.

For example, in Line 352, we have changed the sub-title “The sub-clusters of epithelial cells exhibit elevated stem-like features to promote the aggressiveness of ADC” into “The sub-clusters of epithelial cells in ADC exhibit elevated stem-like features”. In this paragraph, many over-affirmative discriptions such as “exclusively”, “significant”, “overwhelmingly”, “remarkably” have been deleted. From Line 486-493, the conclusion of “Moreover, SLC26A3 could be employed as a marker for the Epi_10_CYSTM1 cluster, aiding in the diagnosis of lymph node metastasis to prevent post-surgical upstaging in ADC patients in the future” have been changed into “our results propose that SLC26A3 might be considered as a diagnostic marker to predict lymph node metastasis in ADC patients”. Similar over-affirmative descriptions and conclusions had also been re-written in the other paragraphs, which has been refered to question (7) above.

(2) This leads to problematic conclusions. For example, the "ADC-specific" Epi_10_CYSTM1 cluster, which is a central focus of the paper, only contains cells from one of the 11 ADC samples and represents only a small fraction of the malignant cells from that sample (Sample 7, Figure 2A). Yet, this population is used to derive SLC26A3 as a potential biomarker. SLC26A3 transcripts were only detected in this small population of cells (none of the other ADC samples), which makes me question the specificity of the IHC staining on the validation cohort.

We sincerely feel grateful for this question. This is a quite important question as it is also pointed out by reviewer#1 in question (8) above. In the revised manuscript, we have already optimized our descriptions and have added detailed explanation for the importance of SLC26A3 in the Discussion section (from Line 802 - 823). We agree that the majority of Epi_10_CYSTM1 cells are derived from sample S7. The fact that we detected this cluster in only one patient may be due to sampling differences and the inherent heterogeneity of tumor specimens. However, the relatively high number of cells in this cluster from one stage III patient suggests its presence in ADC and highlights its potential as a diagnostic marker for staging ADC. To further investigate whether this cluster is generally present in ADC patients, we identified and selected candidate genes, such as SLC26A3, ORM1, and ORM2, as representative markers of this cluster, which demonstrated high specificity (as shown in Fig. 3B). We then performed IHC staining on 56 cases of tissue samples, and the results showed positive expression of these markers in the majority of stage III tumor tissues, confirming the existence of this cell cluster (as shown in Supplementary Fig. 3E). In our revised manuscript, we have included an in-depth discussion of this issue in the seventh paragraph of the Discussion section.

(3) This is compounded by technical aspects of the analysis that hinder interpretation. For example, it is clear that the clustering does not perfectly segregate cell types. In Figures 2B and D, it is evident that C4 and C5 contain mixtures of cell type (eg. half of C4 is EPCAM+/CD3-, the other half EPCAM-/CD3+). These contaminations are carried forward into subclustering and are not addressed. Rather, it is claimed that there is a T cell population that is CD3- and EPCAM+, which does not seem likely.

Thank you for your insightful comment. This important point is also raised by reviewer#1 above. In the revised manuscript, we have reanalyzed our scRNA-seq data and listed the canonical marker genes for cell type annotation. Most importantly, as for T cells and its sub-clustering, we have performed quality control and re-analyzed all data for T cells, with contamination excluded. In the reviesed manuscript, we have added the re-analyzed data for T cells in both Fig. 4 and Supplementary Fig. 4.

**Recommendations for the authors:**

**Reviewer #1 (recommendations for the authors):**
The text would substantially benefit from an editorial revision of language usage.

We sincerely feel grateful for this suggestion. In our revision, we have conducted language editing and carefully rewritten our manuscript. The changes have been clearly marked in the tracked version of the revised manuscript.

**Reviewer #2 (recommendations for the authors):**
(1) Use statistical approaches to claim enrichment/specificity of populations to given groups (ADC, HPV, etc). Analysis packages like Milo for differential abundance testing would be very helpful.

We feel grateful for this suggestion. In our revision, we have performed statistical analyses for all groups of comparison data. Meanwhile, we have rearranged the figures based on these statistical results, for example, Fig. 1F, Fig. 2D, Fig. 4E, Fig. 5D, Fig. 6D, Supplementary Fig. 1A-B, Supplementary Fig. 2A-B, Supplementary Fig. 4A-B, Supplementary Fig. 5A-B, Supplementary Fig. 6A-B.

(2) In the subclustering, consider a round of quality control to ensure that all cells are of the cell type they are claimed to be. Contaminant clusters/cells could be filtered out or reassigned. This could be supplemented with an automated annotation approach using cell-type references.

We feel thankful for this suggestion. As a result, we have provided copies of scripts in the supplementary data to ensure the quality control of cell type annotation.

(3) An explanation for why SLC26A3 is so rare in the scRNA-seq data, but seemingly common in the IHC staining would be helpful. I am concerned about the specificity of the stain.

We apologize for lacking adequate explanation of SLC26A3 and cluster Epi_10_CYSTM1. This is a quite crucial question as it has been listed above in question (8) of reviewer #1 and question (2) of reviewer #2 (public review section). In the revised manuscript, we have added intenstive discussion about this question in the seventh paragraph of Disccusion section (from Line 801 - 822). In fact, because of the heterogeneity among different individuals and different tumor regions even within one sample, Epi_10_CYSTM1 seemed to be derived from only one sample. However, the relatively high number of cells in this cluster from one late-stage (stage IIIC) patient suggests its presence in ADC and highlights its potential as a diagnostic marker for staging ADC. Furthermore, we have identified SLC26A3, ORM1 and ORM2 as specific markers of this cluser and performed IHC staining. With a positive expression of these markers, the existence of this cluster has been indirectly proved (as shown in Fig. 3B).